# Heads up! Large Language Models Can Perform Tasks Without Your Instruction via Selective Attention Head Masking

**Senyu Han** [1 2 3]   **Hongchuan Zeng** [1 2 3]   **Kai Yu** [1 2 3 4]   **Lu Chen** [1 2 3 4]

## Abstract

Large language models (LLMs) consist of numerous Transformer modules, and while the models can perform various functions, it remains an open question of how these modules are combined to elicit distinct inherent functionalities. In this paper, we investigate the modules inside LLMs and demonstrate that, by simply masking or retaining specific attention heads during inference, LLMs can exhibit specific task functionalities without requiring explicit instructions or modifications to the model parameters. Experiments across various models and tasks reveal that LLMs inherently encode "functional pathways", the structured groups of interdependent attention heads that are crucial for executing specific tasks. These pathways not only govern the model's functional behaviors but also enhance parameter efficiency, as suppressing attention heads outside the pathway can improve task performance. The code is available in this repository: https://github.com/OpenDFM/HeadsUp.

## 1. Introduction

Large language models (LLMs) have demonstrated their powerful capabilities across diverse fields. For the most popular Transformer-based (Vaswani, 2017) architecture, the model comprises multiple stacked modules such as attention layers and FFN (Feed-Forward Network) layers. After being trained on large amounts of data, the model can support a wide range of functions by prompting texts containing task instructions. To understand the mechanism of how a single LLM processes multiple tasks, interpreting their inner module functionalities has always been of significant interest to research within the community.

Recent works have demonstrated that built-in transformer modules of LLMs exhibit certain functionality. For example, decoder layers of different depths vary in their effects on the inference and functionality of the model, and existing research probes the importance of layers and their impact on the outputs (Men et al., 2024; Bandarkar et al., 2025). A more fine-grained division of model functional modules is the level of neurons within the FFN layers. Before LLMs became popular, many studies suggested that Transformer-based models store their knowledge in the neurons of FFN layers (Dalvi et al., 2019; Geva et al., 2021). Later research on LLMs has further analyzed and localized the types of knowledge stored within these neurons, such as languages (Zeng et al., 2025; Tan et al., 2024), concepts (Rai & Yao, 2024) and various tasks (Xiao et al., 2024). Apart from the two modules above, many works also study the functional behaviors of attention heads. Although it is generally believed that function-related knowledge is primarily stored in the neurons of FFN layers, there are also considerable works indicating that attention heads play a crucial role in knowledge storage and processing, including long-context retrieval, knowledge editing and more (Wu et al., 2025; Todd et al., 2024; Jin et al., 2024). As these works above provide valuable insights into the model module functionality, most of their methodologies rely on certain tasks or carefully designed prompts, limiting their generalization to other uncovered tasks. Furthermore, these works often require explicit instructions to elicit the corresponding task functionalities of the model, but the reasons and mechanisms behind how a module of the same model supports different functions are still not well explored.

Interestingly, in neuroscience, many similar cases have identified the phenomenon of functional partitioning in the brain (Bertolero et al., 2015; Wig, 2017; Béna & Goodman, 2025), where various functions can be exhibited through different combinations of modules (Diez et al., 2015). Inspired by the modularity in brains, we suppose that the combinations of LLM modules form functionalities inside the model, and we propose a simple yet effective method to detect the task functionalities in LLMs. More specifically, we investigate

---

[1]X-LANCE Lab, School of Computer Science, Shanghai Jiao Tong University, Shanghai, China [2]MoE Key Lab of Artificial Intelligence, Shanghai, China [3]Jiangsu Key Lab of Language Computing, Suzhou, China [4]Suzhou Laboratory, Suzhou, China. Correspondence to: Lu Chen <chenlusz@sjtu.edu.cn>.

*Proceedings of the $42^{nd}$ International Conference on Machine Learning*, Vancouver, Canada. PMLR 267, 2025. Copyright 2025 by the author(s).

the functions of LLM modules by just *removing* these modules from the model. We directly provide the LLM with the task dataset without instructions, and train a 0-1 binary module mask to determine the modules selected for this task. Surprisingly, on the modules of attention heads, this straightforward idea leads us to a marvelous discovery: **by selectively masking attention heads and allowing only certain heads to remain active in LLMs, they can directly exhibit specific task functionalities without the presence of instruction prompts** (Figure 1). We widely confirm this phenomenon across various open-sourced LLMs covering both base and instruct models, various sizes of LLMs ranging from 0.5B to 14B, and various tasks including classification, extraction, generation and more. Our detailed analysis experiments on attention heads suggest that LLMs contain functional *pathways*, which are groups of interdependent attention heads that work together to enable specific functionalities. Such pathways play critical roles when the model performs related functions, and disrupting these dependent attention heads can significantly worsen performance on related tasks. Furthermore, as a de-noising process for the attention heads, if attention heads outside the pathway are reduced or pruned, the model may even exhibit improved performance. Our findings reveal the inherent capabilities of attention heads and locate their specific functional combinations within the model, providing valuable insights into the inner functionalities and behaviors of LLMs.

Our contributions can be summarized as follows:

- We find that masking out certain attention heads can trigger the task functionality in the LLM without using prompts, and we propose a simple learning-based method to detect them (Section 3).
- The selected heads form an interdependent functional pathway on the model, and they have a more critical impact on the model's functionalities (Section 4).
- By scaling down the outputs of heads outside the functional pathway, we can effectively mitigate the negative impact of these heads on the model, and even improve the model's performance on the task (Section 5).

## 2. Related Work

**Discovering functionality in LLMs** Previous studies utilize diverse approaches to locate the modularity or functionality of LLMs. For neurons in FFN layers, the activation function works as a natural judge to determine whether the neuron is important to the input, and many works use the activating frequency (Tan et al., 2024; Zeng et al., 2025) or the output magnitude (Xiao et al., 2024) to partition the model. At the granularity of attention heads, other methods can be applied, such as comparing head logit differences and calculating attention scores (Zheng et al., 2024). Besides these granularity specified methods, many works also use

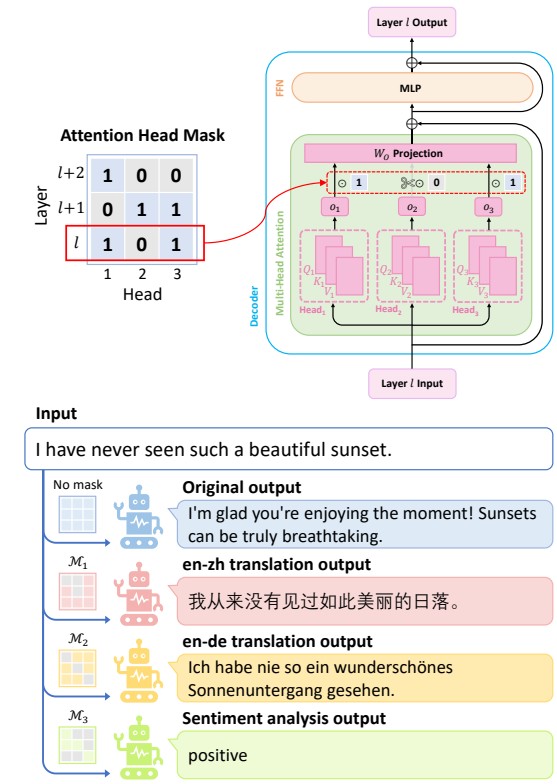

*Figure 1.* By masking and selecting certain attention heads, the model can directly exhibit task functionalities and generate different outputs under the same input.

universal approaches to detect the parameter functionality, like mapping module parameters or outputs to the vocabulary space (Wendler et al., 2024; Feng et al., 2024; Rai & Yao, 2024; Elhelo & Geva, 2024) or comparing parameter changes after task fine-tuning (Zhang et al., 2024; Panigrahi et al., 2023; Xu et al., 2025; Wu et al., 2024b). Particularly, some PEFT-related works also use the strategies of scaling and biasing the output of attention heads (Yin et al., 2024; Wu et al., 2024a), which is similar to our attention mask. In contrast, we only use the discrete scale of 0 or 1 to select the heads, and such unmodified head parameters are better able to reflect the *original* behavior of the model.

**Functionality in attention heads** Attention heads are believed to be capable of performing functions in editing the information flow (Zheng et al., 2024). In smaller pre-trained language models, Jo & Myaeng (2020) evaluate each attention heads on various tasks and finds that heads differ in their influence. For LLMs, Jin et al. (2024) use the gradient-based method to locate "memory heads" and "context heads", and Wu et al. (2025) further locate "retrieval heads" by counting their retrieval scores. Such functionality of attention heads is too generic and task-dependent, making it unsuitable for wide application on other tasks. Other works also manage to locate the important heads in respective of different tasks or knowledge (Todd et al., 2024; Yao et al., 2024), but they

usually require carefully designed prompts to work well. In our work, the training of attention masks allows any ordinary task datasets, and the usage or removal of heads can directly reflect their *inherent* functionalities on the task.

# 3. Attention Head Selection Extensively Triggers Functionality

In this section, we provide our main experiment results of the attention head selection. We first define the notations, methods and settings used in the module selection, then we show the conclusion that attention head selection can extensively trigger functionalities in LLMs.

## 3.1. Preliminaries and Notations

Modern LLMs are commonly decoder-only Transformers, and a decoder layer includes a multi-head attention (MHA) layer and an FFN layer. For an LLM that has a hidden size of $d$, consider the input $\mathbf{X} \in \mathbb{R}^{s \times d}$ with the length $s$ at the decoder layer $l$. Four projection matrices $\mathbf{W}_Q, \mathbf{W}_K, \mathbf{W}_V, \mathbf{W}_O \in \mathbb{R}^{d \times d}$ compute the attention scores and write the processed hidden states $\mathrm{MHA}(\mathbf{X})$ into the residual stream. For the MHA part, suppose the attention layer has $h$ heads[1], and each head has a dim size of $d_{\text{head}} = d/h$. Thus for head $i$, its contribution $\mathbf{Y}^{(i)} \in \mathbb{R}^{s \times d}$ to the MHA output is

$$\mathbf{Y}^{(i)} = \mathrm{softmax}\left( \frac{\mathbf{X}\mathbf{W}_Q^{(i)}(\mathbf{X}\mathbf{W}_K^{(i)})^T}{\sqrt{d_{\text{head}}}} \right) \mathbf{X}\mathbf{W}_V^{(i)}\mathbf{W}_O^{(i)},$$
(1)

where $\mathbf{W}_Q^{(i)}, \mathbf{W}_K^{(i)}, \mathbf{W}_V^{(i)} \in \mathbb{R}^{d \times d_{\text{head}}}$ are submatrices belonging to head $i$, and $\mathbf{W}_O^{(i)} \in \mathbb{R}^{d_{\text{head}} \times d}$ is the "head" submatrix that projects the output of head $i$ to the hidden space. In this way, the final output of MHA can be calculated as $\mathrm{MHA}(\mathbf{X}) = \sum_i^h \mathbf{Y}^{(i)}$.

Using this notation, it's easy to block the output of certain heads by simply setting their contributions to zero. Specifically, for layer $l$ we use a binary mask $\mathbf{m} \in \{0,1\}^h$ to indicate which heads are selected in the model, and the layer MHA output can be written as:

$$\mathrm{MHA}(\mathbf{X}) = \sum_i^h \mathbf{m}_i \mathbf{Y}^{(i)}.$$
(2)

There will be different masks at different layers of the model. For an LLM with $n$ layers, we expect a mask $\mathcal{M} \in \{0,1\}^{n \times h}$ to indicate all the selected attention heads in the model. When applying $\mathcal{M}$ to the model, it can be regarded as combining the selected attention heads as a group

---

[1] Some LLMs have fewer KV heads than Q heads, and the matrices of KV heads are repeated to be equal in size during the runtime. We take $\mathbf{W}_K, \mathbf{W}_V$ as already repeated for simplicity.

in the mask. Using the similar approach, we can also block the output of certain FFN layers, though the mask mostly just simply use all the layers (see Appendix C.3.3). Here, we will only apply masks to attention heads.

## 3.2. Mask Training

For a certain task $t$, We use the training-based method to get its attention head mask $\mathcal{M}$. Specifically, we freeze all the original parameters in the model, and only train a head weight $\mathbf{M}$ that has the same size with $\mathcal{M}$ but in the $\mathbb{R}^{n \times h}$ space. Weights in $\mathbf{M}$ can indicate the importance of heads to the task, and the binary mask $\mathcal{M}$ can be obtained by the sigmoid function $\mathcal{M} = \mathbb{I}(\sigma(\mathbf{M}) \geq 0.5)$. Inspired by Fang et al. (2024) and Csordás et al. (2021), we introduce the Gumbel trick (Gumbel, 1954) into the sigmoid function to encourage $\sigma(\mathbf{M})$ to converge towards 0 or 1 in the training:

$$\mathcal{M} = \mathbb{I}\left( \sigma\left( \frac{\mathbf{M} + \mathbf{G}}{\tau} \right) \geq 0.5 \right),$$
(3)

where $\mathbf{G} = \log(-\log \mathcal{E}), \epsilon_{ij} \in \mathcal{E}, \epsilon_{ij} \sim U(0,1)$ is the left-tail Gumbel noise matrix and $\tau$ is the temperature. Smaller $\tau$ amplifies the impact of noise while also encouraging the weights $\mathbf{M}$ to shift toward the positive or negative extremes. Using the reparametrization trick, we can get a discrete but still differentiable mask $\mathcal{M}$ during the training.

Just like fine-tuning an ordinary language model, it's straightforward to train an attention head mask: for a sample $(x, y)$ in the task dataset $\mathcal{D}_t$, we directly take the $x \| y$ as input without any task instructions and optimize the head weight $\mathbf{M}$ on the output sequence $y$ to minimize the loss. Note that the weight $\mathbf{M}$ is initialized with a positive value so that initial $\sigma(\mathbf{M}) \approx \mathbf{1}$, and the training process is like removing attention heads from the full model. After training, we remove the Gumbel noise from (3) and use $\mathcal{M} = \mathbb{I}(\sigma(\mathbf{M}) \geq 0.5)$ as the final mask for inference.

## 3.3. Experiment Setting

**Models**  To verify the generalization of this attention selection strategy, we experiment with different types and sizes of LLMs as follows:

- Llama 3.1 series (Dubey et al., 2024): 8B, 8B-instruct
- Qwen2.5 series (Team, 2024b): 0.5B-instruct, 1.5B-instruct, 3B-instruct, 7B, 7B-instruct, 14B-instruct
- Other series of LLMs: Phi-3.5-mini-instruct (Adina-Tru, 2024, with 3.8B params), Mistral-7B-Instruct-v0.3 (Jiang et al., 2023)

Details of their size and attention head settings are listed in Appendix A. We primarily conduct experiments using two 8B Llama 3.1 models, while other models are mainly used

to validate the effectiveness of the attention head mask. By convenience, we refer to the 8B-instruct version of Llama 3.1 as Llama3.1$_{8B\text{-inst}}$ in this paper, and other models follow similar patterns. We use LM Transparency Tool (Tufanov et al., 2024) to probe the logit outputs inside the models.

**Tasks and datasets**  We train the head mask on two types of tasks: simple tasks and machine translation. Simple tasks are taken from the work by Todd et al. (2024), including 35 datasets of relatively simple tasks like generating antonyms and extracting items. Machine translation uses the XNLI (Conneau et al., 2018) parallel corpus for 15 languages, and we choose English, Chinese, French, Spanish, German, Russian and Arabic (en, zh, fr, es, de, ru, ar) for translation. These 7 languages can be combined into 42 translation pairs for a broad generalization of the head mask. Both types of tasks have the characteristic that **models can hardly generate the desired output solely based on the input unless they are explicitly instructed on the task to be performed.** For each dataset, we select 100 samples as the evaluation set, and the remaining samples are used for training. Details of these datasets are listed in Appendix A.

### 3.4. Experiment Results

Following the paradigm defined above, we train head masks for different tasks on various models. Our finding is: these masks extensively work on all the tasks, and such attention head combinations trigger the functionalities on both pre-trained and instruction fine-tuned models.

**Evaluation results**  We report the perplexity (PPL) and ROUGE-L (Lin, 2004) of Llama3.1$_{8B\text{-inst}}$ and Llama3.1$_{8B}$ on 6 English translation tasks, as shown in Table 1. With the mask $\mathcal{M}$, by removing 15%-20% attention heads from the model, both models can reach better or comparable scores than the full models with explicit translation instructions (e.g. "Translate into Chinese:"). As an ablation, randomly selecting an equal number of attention heads results in a higher PPL than the original full model, indicating that the learned $\mathcal{M}$ effectively covers the attention heads necessary for the task. Additionally, we attempt to swap the masks between pre-trained and instruction models for the same task, finding that their PPL also decreases compared to the original full model. Appendix C.1 presents the experiment results on other translation pairs and models. Except for the relatively poor performance on Qwen2.5$_{0.5B\text{-inst}}$, other models consistently yield similar conclusions.

Simple tasks also show the similar phenomenon. We select 6 representative tasks out of 35 in Table 2: sentiment (*classification*), antonym (*semantic*), capitalize_first_letter (*pattern*), product-company (*knowledge*), conll2003_location (*recognition*) and fruit_v_animal_3 (*logic*). In this generation scenario, the original full model can hardly generate the desired output without task instructions. Instead, using

*Table 1.* Numbers of attention heads selected in the masks, and the metrics of translation tasks from English to other 6 languages. "Original" uses no instructions nor masks, and "w/ instruction" adds explicit translation prompts to the original model. "Random $\mathcal{M}$" shuffles the task mask and has the equal number of heads. **Bold** and underline indicate the best and second-best scores.

| Method | en-zh | en-fr | en-es | en-de | en-ru | en-ar |
|---|---|---|---|---|---|---|
| **Llama3.1$_{8B\text{-inst}}$** | | | | | | |
| - # of heads in $\mathcal{M}$ | 832 | 860 | 846 | 862 | 828 | 837 |
| **PPL** ($\downarrow$) | | | | | | |
| - Original | 2.643 | 2.172 | 2.319 | 2.314 | 2.327 | 2.730 |
| - w/ instruction | 1.091 | 0.619 | 0.550 | 0.627 | **0.954** | **1.303** |
| - w/ random $\mathcal{M}$ | 2.909 | 2.370 | 2.762 | 2.482 | 2.677 | 3.204 |
| - w/ $\mathcal{M}$ | **1.017** | **0.552** | **0.504** | **0.580** | 1.055 | 1.322 |
| - w/ $\mathcal{M}$ of Llama3.1$_{8B}$ | 2.269 | 1.114 | 1.061 | 1.612 | 2.105 | 2.381 |
| **ROUGE-L** ($\uparrow$) | | | | | | |
| - Original | 0.020 | 0.082 | 0.069 | 0.069 | 0.045 | 0.029 |
| - w/ instruction | 0.539 | 0.541 | 0.599 | 0.576 | **0.491** | 0.414 |
| - w/ $\mathcal{M}$ | **0.589** | **0.711** | **0.748** | **0.677** | 0.401 | **0.427** |
| - w/ $\mathcal{M}$ of Llama3.1$_{8B}$ | 0.019 | 0.204 | 0.241 | 0.099 | 0.046 | 0.045 |
| **Llama3.1$_{8B}$** | | | | | | |
| - # of heads in $\mathcal{M}$ | 819 | 809 | 811 | 816 | 795 | 821 |
| **PPL** ($\downarrow$) | | | | | | |
| - Original | 2.308 | 1.672 | 1.853 | 1.780 | 1.824 | 2.434 |
| - w/ instruction | 1.901 | 1.240 | 1.350 | 1.258 | 1.573 | 1.945 |
| - w/ random $\mathcal{M}$ | 2.703 | 2.329 | 2.337 | 2.133 | 2.377 | 3.251 |
| - w/ $\mathcal{M}$ | **1.158** | **0.585** | **0.577** | **0.660** | **1.142** | **1.867** |
| - w/ $\mathcal{M}$ of Llama3.1$_{8B\text{-inst}}$ | 2.213 | 1.512 | 1.515 | 1.480 | 1.754 | 2.393 |
| **ROUGE-L** ($\uparrow$) | | | | | | |
| - Original | 0.017 | 0.072 | 0.067 | 0.066 | 0.045 | 0.030 |
| - w/ instruction | 0.017 | 0.084 | 0.080 | 0.076 | 0.054 | 0.029 |
| - w/ $\mathcal{M}$ | **0.424** | **0.658** | **0.657** | **0.606** | **0.179** | **0.067** |
| - w/ $\mathcal{M}$ of Llama3.1$_{8B\text{-inst}}$ | 0.016 | 0.071 | 0.074 | 0.076 | 0.037 | 0.030 |

partial attention heads selected by $\mathcal{M}$ only can trigger the model functionality, and significantly improve the accuracy of these tasks. The results on other tasks and experiments on Qwen2.5$_{7B\text{-inst}}$, Qwen2.5$_{7B}$ are listed in Appendix C.1, and on simple tasks the pre-trained model Qwen2.5$_{7B}$ also exhibits well functionality.

Besides, we found that the model's few-shot in-context learning functionality can also be enhanced by applying the head mask, as shown in Table 3. For the 6 representative simple tasks, we constructed 5-shot demonstration samples (without instructions) for each task. The masks trained using these samples generally outperform the original model with instructional 5-shot samples. Furthermore, we mix the 5-shot data from 32 out of 35 simple tasks to form a 5-shot multi-task *hybrid* dataset. After training the mask on this hybrid dataset, the model demonstrates improved few-shot in-context learning performance across various tasks, even including those not seen in the training data. This suggests that the attention head mask can also promote the model's abstract functionality of in-context learning.

**Larger model shows better head functionality**  As a part of scaling validation, when using the head mask to trigger translation functionalities, we find that larger LLMs tend to have better results. In Table 4, we show the genera-

*Table 2.* Numbers of attention heads selected in the masks, and the accuracy of 6 representative simple tasks. Using the same notation as in the Table 1.

| Method | senti-ment | anto-nym | capitalize first letter | product-company | conll2003 location | fruit v animal |
|---|---|---|---|---|---|---|
| **Llama3.1$_{8B\text{-inst}}$** | | | | | | |
| - # of heads in $\mathcal{M}$ | 745 | 834 | 774 | 754 | 787 | 827 |
| **Accuracy (↑)** | | | | | | |
| - Original | 0.01 | 0.12 | 0.15 | 0.06 | 0.00 | 0.00 |
| - w/ instruction | 0.59 | 0.30 | 0.97 | 0.73 | 0.41 | 0.51 |
| - w/ $\mathcal{M}$ | **0.86** | **0.76** | **1.00** | **0.83** | **0.92** | **0.98** |
| - w/ $\mathcal{M}$ of Llama3.1$_{8B}$ | 0.64 | 0.75 | 0.98 | 0.78 | 0.75 | 0.87 |
| **Llama3.1$_{8B}$** | | | | | | |
| - # of heads in $\mathcal{M}$ | 679 | 726 | 678 | 674 | 670 | 786 |
| **Accuracy (↑)** | | | | | | |
| - Original | 0.00 | 0.13 | 0.00 | 0.07 | 0.07 | 0.00 |
| - w/ instruction | **0.91** | 0.66 | **1.00** | 0.75 | **0.79** | **0.99** |
| - w/ $\mathcal{M}$ | 0.80 | 0.66 | 0.99 | **0.76** | **0.79** | 0.97 |
| - w/ $\mathcal{M}$ of Llama3.1$_{8B\text{-inst}}$ | 0.00 | 0.11 | 0.06 | 0.35 | 0.37 | 0.43 |

*Table 3.* Accuracy of 6 representative simple tasks in the 5-shot scenario on Llama3.1$_{8B\text{-inst}}$. *Italic* indicates that the corresponding task is not included in the hybrid dataset (unseen during the training).

| Method | senti-ment | anto-nym | capitalize first letter | product-company | conll2003 location | fruit v animal |
|---|---|---|---|---|---|---|
| Original, 5-shot | 0.96 | 0.56 | 0.95 | 0.77 | 0.61 | 0.98 |
| w/ instruction, 5-shot | 0.97 | 0.67 | **1.00** | 0.84 | 0.72 | **1.00** |
| w/ $\mathcal{M}$, 5-shot | **0.99** | **0.73** | 0.99 | **0.88** | **0.95** | **1.00** |
| w/ $\mathcal{M}$ of hybrid, 5-shot | *0.98* | 0.69 | **1.00** | *0.88* | *0.82* | **1.00** |

tion examples of four translation pairs on Qwen2.5 series. Smaller models suffer from difficulties in triggering target functionality (e.g. Qwen2.5$_{0.5B\text{-inst}}$ on en-es directly takes the input as instruction), aligning to the target language (e.g. Qwen2.5$_{0.5B\text{-inst}}$ on en-de), and accurately translating the sentence. From the perspective of attention heads, as larger models usually have more attention heads, there are more combinations of heads available to correspond to more functionalities. On the other hand, Qwen2.5$_{0.5B\text{-inst}}$ and Qwen2.5$_{1.5B\text{-inst}}$ have the same amount of attention heads, but the latter has larger hidden size and more parameters in FFN layers, which may work better to process the functionality from incomplete MHA output. A more illustrative example is shown in Table 4a and 4b, where we probe the last layer's output logits on the last token of the input. When using the head mask of the translation task en-de, Qwen2.5$_{0.5B\text{-inst}}$ takes "I" as the top-1 token, but German words "Es" *(It)*, the expected "Ich" *(I)* and Russian "Я" *(I)* are also in top tokens. In contrast, with the explicit instruction, the model can correctly predict the token "Ich" as the top-1 token. This suggests that the small models may have difficulty in distinguishing and triggering the functions of different attention heads.

**Pre-trained models also possess functionalities** Even though pretrained models (base models without instruction fine-tuning) generally perform worse than instruction fine-tuned models on both types of tasks, we observe that they still exhibit certain functionalities, especially in simple tasks.

*Table 4.* **Left:** After applying the head masks, the generated results of the English input *"I have never seen such a beautiful sunset."* across different sizes of models. We use green to highlight the correct translations, yellow for partially correct translations, and red for incorrect translations. **Right:** Top output tokens and their logits on Qwen2.5$_{0.5B\text{-inst}}$. Upper (a) is with the $\mathcal{M}$ of task en-de, and lower (b) is with the explicit translation instruction.

| Model & Task | Prediction |
|---|---|
| **Qwen2.5$_{0.5B\text{-inst}}$** | |
| - en-zh | 我从未见过如此美丽的日落。 |
| - en-fr | Je n'ai jamais vu tellement une belle lueur de ciel. |
| - en-es | I'm sorry, but I can't provide the exact description of a sunset ... |
| - en-de | I've never seen such a beautiful sunset. |
| **Qwen2.5$_{1.5B\text{-inst}}$** | |
| - en-zh | 我从未见过如此美丽的日落。 |
| - en-fr | Je n'ai jamais vu un coucher de soleil aussi beau. |
| - en-es | He nunca había visto un solsticio tan hermoso. |
| - en-de | I can't say I've ever witnessed one like that before. |
| **Qwen2.5$_{3B\text{-inst}}$** | |
| - en-zh | 我从没见过如此美丽的日落。 |
| - en-fr | J'ai jamais vu un tel coucher de soleil si beau. |
| - en-es | Nunca había visto una puesta de sol tan hermosa. |
| - en-de | Ich habe nie so ein wunderschönes Sonnenuntergang gesehen. |

(a) w/ $\mathcal{M}$

(b) w/ inst

*Table 5.* Jaccard similarities of head masks on 12 tasks between Llama3.1$_{8B\text{-inst}}$ and Llama3.1$_{8B}$. "Random" calculates the math expectation of Jaccard similarity on two randomly shuffled masks.

| Task | en-zh | en-fr | en-es | en-de | en-ru | en-ar |
|---|---|---|---|---|---|---|
| Similarity | 0.766 | 0.766 | 0.797 | 0.802 | 0.740 | 0.762 |
| Random | 0.675 | 0.687 | 0.679 | 0.693 | 0.656 | 0.680 |

| Task | sentiment | antonym | capitalize first letter | product-company | conll2003 location | fruit v animal |
|---|---|---|---|---|---|---|
| Similarity | 0.591 | 0.707 | 0.612 | 0.634 | 0.615 | 0.707 |
| Random | 0.531 | 0.610 | 0.545 | 0.533 | 0.547 | 0.649 |

This makes us believe that the pre-trained models have already learned some task-related knowledge in the attention heads during the next-token prediction pre-training. As swapping the head masks can also trigger certain functionalities in some cases, we further investigate the similarity of the selected attention heads between the pre-trained and instruction models. For two masks of Llama3.1$_{8B\text{-inst}}$ and Llama3.1$_{8B}$ on the same task, we calculate the Jaccard similarity of the selected attention heads. Table 5 shows that the mask similarity is generally higher than the random expectation, but still lower than identical 1.0, as masks of the instruction model tend to use more heads. We suppose that the instruction model inherits most of the attention head functionalities from the pre-trained model, and also introduces new functionalities that are not present in the pre-trained model during the instruction fine-tuning.

# 4. Attention Head Selection Forms Functional Pathways

In this section, we further investigate how the combination of selected attention heads triggers the task functionality in the LLM. We first define the *functional pathway* as a group of interdependent attention heads that work together to enable specific functionalities. In the following experi-

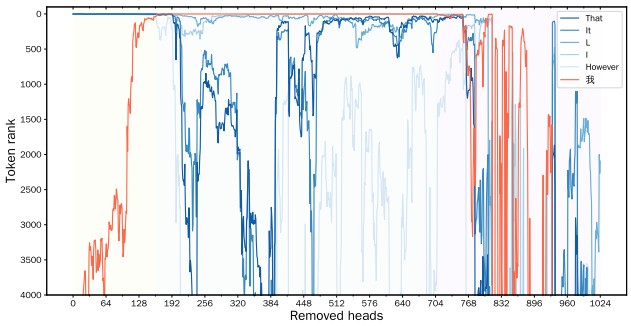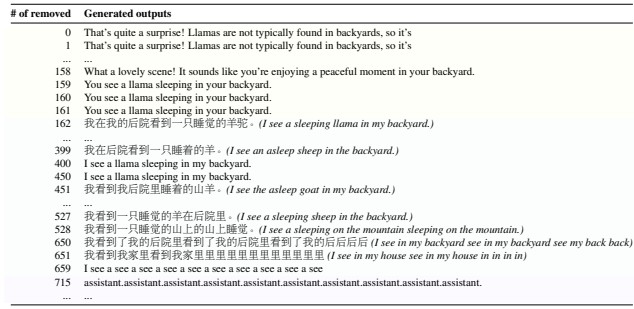

*Figure 2.* Prompt Llama3.1**8B-inst** with *"I see a llama sleeping in my backyard."*, and remove attention heads from the full model by head weight of en-zh translation task in ascending order. **Left:** With the removal of attention heads, changes of token ranks in the first generated token. Blue lines indicate the original top-5 tokens and the red line indicates the target token "我" *(I)* for translation. **Right:** Generated outputs of the model after removing different amounts of attention heads.

ments, We find that attention heads selected by the mask form such pathways in the model, and disrupting or selectively masking these heads significantly alters the model's task performance. We further reveal that these heads tend to shift the model's original behavior towards the task functionality, and such a pathway effect is unlikely caused by paying more attention to certain tokens.

**Attention heads form critical, interdependent functional pathways inside the model** We first want to know how masking certain attention heads can switch the model's behavior from instruction-following towards task functionality. One way to probe the behavior changes is progressively removing attention heads one by one from the full model. On model Llama3.1**8B-inst** and the en-zh translation task, we start with the attention head that has the minimum weight in **M**, remove heads in the weight ascending order, and check the changes in the generated responses. As shown in Figure 2, the model experiences multiple stages in behaviors:

- *Stage 1:* The model can still preserve the functionality as an instruction model, but when approximately 100 attention heads are removed, the rank of the target translation token (red line) experiences a drastic increase. This surge indicates that the model's translation functionality begins to emerge.

- *Stage 2:* After removing 158 attention heads, the model undergoes a *rapid* transition, and starting from the removal of the 162nd head, it shifts from its original instruction-following behavior to a translation behavior. This task functionality begins to dominate, and further removal of attention heads causes the rankings of initial top tokens (blue lines) to drop rapidly.

- *Stage 3:* As more attention heads are removed, after 528 heads, the model's language processing ability begins to lose, and the generated tokens will become meaningless in the end. This is expected as the model has lost the majority of its attention heads.

*Table 6.* PPL of explicit translation tasks from English to other languages when applying different masks on Llama3.1**8B-inst**. We remove attention heads from the full mask **1** that: "all" - used by all 42 translation tasks, "en" - used by 12 translation tasks related to English, and "unused" - not used by any translation tasks.

| Method | # of heads | en-zh | en-fr | en-es | en-de | en-ru | en-ar |
|---|---|---|---|---|---|---|---|
| Instruction | 1024 | 1.091 | 0.619 | 0.550 | 0.627 | 0.954 | 1.303 |
| w/ $1_{\backslash\text{all}}$ | 847 | 4.322 | 4.129 | 4.149 | 4.406 | 5.008 | 5.441 |
| w/ random $1_{\backslash\text{all}}$ | 847 | 2.122 | 0.792 | 0.864 | 1.248 | 1.237 | 2.578 |
| w/ $1_{\backslash\text{en}}$ | 741 | 7.029 | 6.137 | 6.396 | 6.511 | 6.903 | 7.011 |
| w/ random $1_{\backslash\text{en}}$ | 741 | 3.325 | 1.480 | 2.136 | 1.540 | 2.619 | 2.972 |
| w/ $1_{\backslash\text{unused}}$ | 977 | **1.064** | **0.577** | **0.497** | **0.585** | **0.924** | **1.269** |
| w/ random $1_{\backslash\text{unused}}$ | 977 | 1.112 | 0.688 | 0.705 | 0.603 | 1.020 | 1.369 |

Specifically, we observe that when attention heads are removed, there are numerous abrupt, drastic changes in token ranks. If the attention heads worked independently, then firstly removing those heads that significantly promote the target token rank would accelerate the behavior shift of the model. However, this influential removal order actually delays the functionality switch (see more in Appendix C.3). This suggests that the behavior changes are not caused by removing certain independent attention heads, but are built upon the removal of all previously removed attention heads.

Considering such dependency between attention heads, if we take the selected attention heads as nodes and connect them with edges, then we can get an interdependent functional pathway that influence model behaviors. To verify the impact of such head pathways on model functionality, we initialize the head weights **M** with negative values (so that initial $\sigma(\mathbf{M}) \approx \mathbf{0}$) when training the mask $\mathcal{M}$. This approach yields us masks with fewer and more task-specific attention heads while slightly damaging language capability. We train these masks on 42 pairs of translation tasks, identify the heads that are commonly used or unused, remove these common heads from the model and evaluate PPL on translation tasks. As these common heads are used in every translation task, we can consider them as key components of the translation function pathway. Within our expectation,

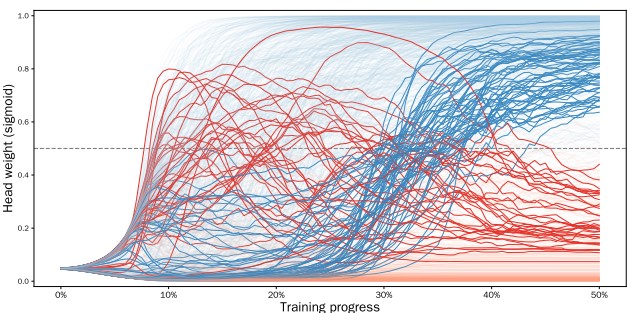
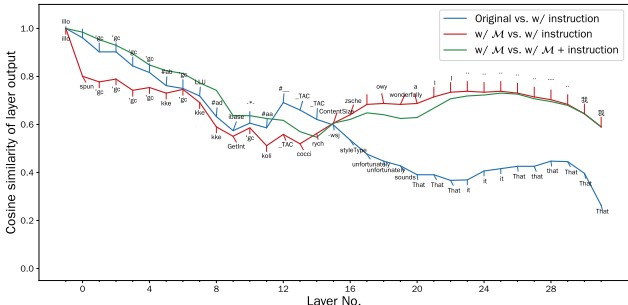

*Figure 3.* Changes of head weights in the mask training progress. Blue lines indicate the heads that are selected in the final mask, and red lines indicate the heads that are not selected. We highlight the heads that undergo dependent changes during the training.

*Figure 4.* Cosine similarities of hidden states between different settings after each layer. Dark lines indicate the outputs of decoder (also FFN) layers and light lines indicate the outputs of attention layers. We also project the decoder hidden states of Original and w/ $\mathcal{M}$ into the vocabulary space and label their top-1 tokens.

in Table 6 we find that blocking the key translation-related attention heads in the pathway can lead to a severe performance drop on the translation tasks.[2] This demonstrates that combined attention heads inside the task pathway significantly influences the related functionality of the model.

We further explain the point that heads inside the task pathway do not just independently retain task-related functions. A piece of illustrative evidence in Figure 3 shows that attention heads in the pathway are dependent on each other, where we probe into the changes in head weights during the mask training. We notice that many unused heads in the final $\mathcal{M}$ initially have been temporarily selected during the training (sigmoid weight greater than 0.5), but as more other heads join the pathway during the training progress, these heads are eventually ablated from the pathway. A similar phenomenon can be also observed in some heads that are selected in the final mask, as they are not selected in the early training stage but are added to the pathway later. From this perspective, such functionality exhibited by the combination of attention heads is not independently triggered by the individual attention heads being combined, but is instead triggered through functional pathways formed by their interdependence.

**Attention head masks attempt to reconstruct the task functionality** We further investigate why the functional pathway formed by attention heads can trigger the task functionality without explicit instructions. Intuitively, if the output of the model using head mask $\mathcal{M}$ on the last token is similar to that of the model with instructions, it can be considered that the pathway formed by the mask is attempting to approximate the information flow required to execute the function. Again, we prompt Llama3.1$_{\text{8B-inst}}$ with the sentence *"I see a llama sleeping in my backyard."*, and calculate the cosine similarities of layer hidden state outputs between different model and input settings, as shown in Fig-

ure 4. We use three settings for comparison: (Blue) Original vs. w/ instruction ("Translate into Chinese"), which uses the same model but different functions; (Red) w/ $\mathcal{M}$ vs. w/ instruction, which performs the same function but differs in model; and (Green) w/ $\mathcal{M}$ vs. w/ $\mathcal{M}$ + instruction, which performs the same function in the same model. Compared with the original model without instructions nor masks, the FFN hidden state outputs of the masked model gain *higher similarity* with the unmasked instructed model in the later layers. It's easy to understand that LLM often performs language understanding and semantic processing at shallower layers, while more complex task-related inferences are carried out at deeper layers (Bandarkar et al., 2025; Wendler et al., 2024), so consequently the model with $\mathcal{M}$ demonstrates an advantage in similarity only in the latter half of the model. Meanwhile, considering the inherent differences in the instructions given to the model inputs, even having the same model perform the same function, the layer outputs still cannot always maintain a high level of similarity at every layer (green line). Compared with it, the advantage of the masked model over the original model in later layer output similarities indicates that: as the layer goes deeper, the combined attention heads are attempting to reconstruct the functional information flow of the model.

In the same scenario depicted in Figure 2, we also calculate the similarities of layer outputs during this one-by-one head removal and present their changes in Figure 5. We find that as attention heads are gradually removed, the similarity in deeper decoder layers between the masked model and instructed model increases, and the value becomes stable once the masked model fully exhibits the target translation functionality (after removing 192 heads). This increase in similarity indicates that the remaining attention heads are reconstructing the task functionality during the process of unveiling the function pathway. In contrast, the similarity of the shallower layer outputs slightly decreases as more attention heads are removed. This can be explained since

---

[2]Surprisingly, removing unused heads results in slight improvements. This enlightens us to further utilize it in Section 5.

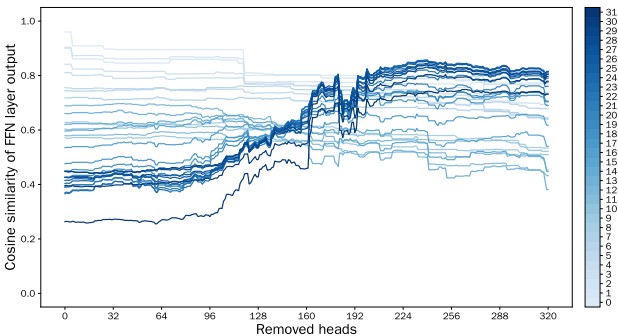

*Figure 5.* Remove attention heads from the full model by weight ascending order, and the cosine similarities of hidden states between w/ instruction and w/ $\mathcal{M}$ after each decoder layer. Darker lines indicate later layers (closer to the model output).

the shallower layers have not yet processed much about functionality, and removing attention heads reduces their capacity to handle language and semantic processing.

**Attention head mask is not selecting attention scores**
Although the attention head selection involves the model's attention mechanism, we did not explore how this selection affects token attention scores in the previous discussion. Normally, in the attention layer, the model attends to the preceding tokens with varying attention scores. However, since our attention selection is independent of instructions, the model's attention will not focus on nonexistent instruction tokens. Especially in some simple tasks, the model input is just a single token, and there is little space for the attention mechanism to work around. From this perspective, such functionality is unlikely caused by simply selecting attention heads that are more focused on the target.

We verify this hypothesis by calculating the average attention score distribution of heads on 4 extractive simple tasks on Llama3.1$_{8B\text{-inst}}$ as they have explicit target tokens in the inputs. The input to these tasks is a set of words, and the objective is to extract the target one that is different from the others. As shown in Table 7, for the original model without task instructions, the model does not pay significant attention to the target word, and as expected, the instructed model has higher attention scores on the target word. However, if we separately calculate the average attention scores of heads in and out of the task mask $\mathcal{M}$, we find that the selected attention heads do not necessarily have higher attention scores (even lower in 3 out of 4 tasks) on the target word. This indicates that the head selection is not based on the attention scores of the instructed model. Even in the masked model, attentions to the target word are further promoted, but the unused heads still have high attention scores on the target word, *notwithstanding* they are not involved in the model inference. Such anomaly in attention scores suggests that the selection of attention heads functions more as influencing the overall behavior of the model. This is consistent

with the previous findings that the selected attention heads are not independently triggering the functionality, but are forming pathways to activate the model.

# 5. Attention Head Selection Potentially Improves Model Performance

In Table 6, we notice that removing translation-unrelated attention heads can decrease the perplexity of the model on the translation task. This suggests that the model may inherently possess a better ability to perform specific tasks, but this ability is suppressed by the noise of other attention heads outside the pathway to better handle a broader range of tasks. To measure the influence of unselected heads to the task, we introduce a scaling factor $\alpha \in [0, 1]$ on the outputs of attention heads outside $\mathcal{M}$, and in this case the head mask can be written as $\mathcal{M} = \mathbb{I}(\sigma(\mathbf{M}) \geq 0.5) + \alpha\mathbb{I}(\sigma(\mathbf{M}) < 0.5)$. We conduct a grid search on 6 explicitly instructed translation tasks of XNLI with Llama3.1$_{8B\text{-inst}}$. As shown in Figure 6, it can be observed that using masks to remove attention heads outside the pathway ($\alpha = 0$) consistently results in lower PPL than using all attention heads ($\alpha = 1$). Meanwhile, this scaling method is not a simple linear interpolation between the full model and the zero-masked model, but a convex combination of the two: using a small $\alpha$ around 0.2 or 0.3 can achieve a lower PPL. To further verify the performance improvement of the model in the generation scenario, we conduct evaluations using the additional translation dataset IWSLT2017 (Cettolo et al., 2017). Under the condition of providing the same translation instructions, we compare the score differences between the original model and the model with the scaled $\mathcal{M}$ applied. As shown in Table 8, using $\mathcal{M}$ consistently improves the model's BLEU scores on these translation tasks, and applying a small scaler $\alpha$ can offer further slight enhancements. From this perspective, selecting attention heads can better showcase the model's inherent but hindered capabilities, potentially improving its performance.

*Table 7.* Average attention scores of attention heads on the target word. These tasks require selecting 1 word that differs from the others among 3 words (so the uniform score is 0.33). We calculate the attention score of each attention head on these words for the last token of the input, normalize it, and then sum up the attention on all tokens of the target word as the values below.

| Method | adjective v verb | object v concept | verb v adjective | fruit v animal |
|---|---|---|---|---|
| Original, all heads | 0.352 | 0.306 | 0.314 | 0.347 |
| w/ instruction, all heads | 0.409 | 0.363 | 0.345 | 0.468 |
| w/ instruction, heads in $\mathcal{M}$ | 0.408 | 0.356 | **0.353** | 0.462 |
| w/ instruction, heads in $1 - \mathcal{M}$ | **0.413** | **0.386** | 0.316 | **0.494** |
| w/ $\mathcal{M}$, heads in $\mathcal{M}$ | 0.462 | 0.471 | **0.426** | 0.566 |
| w/ $\mathcal{M}$, heads in $1 - \mathcal{M}$ | **0.484** | **0.537** | 0.379 | **0.607** |

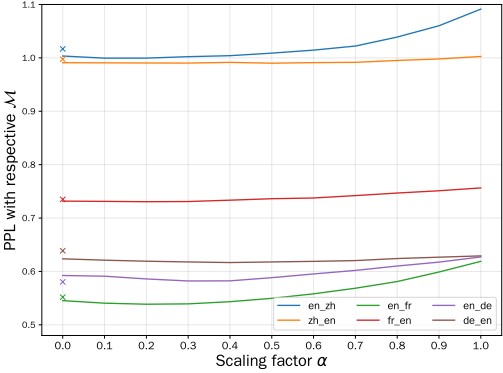

*Figure 6.* PPL of translation tasks when using both explicit instructions and task masks, with different scaling factors $\alpha$. Mark "×" points to the PPL of the model without instructions.

*Table 8.* BLEU scores for explicitly instructed translation pairs on IWSLT2017. Each scaled translation pair uses its own mask with the best scaling factor $\alpha$ shown in Figure 6.

| Method | en-zh | en-fr | en-de | zh-en | fr-en | de-en |
|---|---|---|---|---|---|---|
| Instructed | 0.322 | 0.699 | 0.657 | 0.604 | 0.739 | 0.716 |
| w/ $\mathcal{M}$ | 0.347 | 0.716 | 0.674 | 0.610 | **0.743** | **0.720** |
| w/ scaled $\mathcal{M}$ | **0.349** | **0.717** | **0.678** | **0.611** | **0.743** | 0.719 |

## 6. Conclusion

In this paper, we discover that LLMs can directly exhibit functionality solely through combining attention heads, and we propose a simple method to identify intrinsic functional pathways within the model. By focusing on the model's inherent functionality, we conduct a detailed investigation into its mechanisms, contributing to the interpretability of attention heads and the phenomenon of functional partitioning in LLMs. In the future, we plan to extend similar research to structures such as FFNs and MoE models with the expectation of constructing a comprehensive functional pathway for the model.

## Acknowledgments

This work is funded by the China NSFC Projects (92370206, U23B2057, 62120106006) and Shanghai Municipal Science and Technology Major Project (2021SHZDZX0102).

## Impact Statement

This paper advances the interpretability of large language models, aiming to make their internal reasoning processes more transparent and understandable to users and developers. By improving interpretability, we contribute to more accountable and trustworthy AI systems. There are other many potential societal consequences of our work, none which we feel must be specifically highlighted here.

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

## A. Models & Datasets

In this section, we introduce the models and datasets used in our experiments. Models are selected from various open-sourced LLMs, as listed in Table 9. For translation tasks, we use XNLI (Conneau et al., 2018) for training and main evaluation, and IWSLT2017 (Cettolo et al., 2017) for additional evaluation in Section 5. XNLI contains 10000 samples for each language pair, and we use 100 samples for evaluation and the remaining 9900 for mask training. IWSLT2017 contains 8549 samples for English-Chinese pair, 8597 samples for English-French pair, and 8079 samples for English-German pair. We use all the samples for the evaluation reported in Table 8. For simple tasks, we choose 35 datasets in the Function Vector work by Todd et al. (2024). In each task, we choose out 100 samples for evaluation, and the remaining samples are used for the fixed-step mask training. The details of these datasets and our instruction prompts (for w/ instruction evaluations) are listed in Table 10.

## B. Experimental Details

All experiments are conducted on a NVIDIA A800 80GB GPU. We summarize the hyper-parameters used in the mask training in Table 11. In particular, for memory reasons, the batch size on translation tasks of Qwen2.5$_{14B\text{-}inst}$ is set to 8 and its training steps are 12370. Other hyper-parameters remain the same. For the evaluation of simple tasks, we generate 10 new tokens and check whether the response contains the correct answer, in case the instruct models may generate some leading words (like "the answer is:"). But for some tasks like capitalizing letters and extracting items, we only check whether the response starts with the correct answer. You may check more implementation details in our public code repository.

## C. More Experiment Results

In this section, we provide more experiment results to support the findings in the main text.

### C.1. Full Evaluation Results on All Models

As complementary to the results in Section 3, we provide the full evaluation results on all models in Table 12-35 (translation PPL Table 12-21, translation ROUGE-L Table 22-31, simple task accuracy Table 32-35). From these complete results, we also find some other phenomena:

- Qwen2.5$_{14B\text{-}inst}$ does not give the dominant ROUGE-L performance in the instructed translation task, although it is the largest model in our experiments. We examine the translation outputs of Qwen2.5$_{14B\text{-}inst}$ and Qwen2.5$_{7B\text{-}inst}$, finding that the 14B model tends to generate more explanation words than the 7B model

*Table 9.* LLMs used in this paper. We mainly report their attention layer configurations.

| Model name | Notation | Params | Layers | Heads | KV heads | Total heads |
|---|---|---|---|---|---|---|
| Llama-3.1-8B-Instruct | Llama3.1$_{8B\text{-}inst}$ | 8.03B | 32 | 32 | 8 | 1024 |
| Llama-3.1-8B | Llama3.1$_{8B}$ | 8.03B | 32 | 32 | 8 | 1024 |
| Qwen2.5-0.5B-Instruct | Qwen2.5$_{0.5B\text{-}inst}$ | 494M | 24 | 14 | 2 | 336 |
| Qwen2.5-1.5B-Instruct | Qwen2.5$_{1.5B\text{-}inst}$ | 1.54B | 28 | 12 | 2 | 336 |
| Qwen2.5-3B-Instruct | Qwen2.5$_{3B\text{-}inst}$ | 3.09B | 36 | 16 | 2 | 576 |
| Qwen2.5-7B-Instruct | Qwen2.5$_{7B\text{-}inst}$ | 7.62B | 28 | 28 | 4 | 784 |
| Qwen2.5-7B | Qwen2.5$_{7B}$ | 7.62B | 28 | 28 | 4 | 784 |
| Qwen2.5-14B-Instruct | Qwen2.5$_{14B\text{-}inst}$ | 14.8B | 48 | 40 | 8 | 1920 |
| Phi-3.5-mini-instruct | Phi3.5$_{mini\text{-}inst}$ | 3.82B | 32 | 32 | 32 | 1024 |
| Mistral-7B-Instruct-v0.3 | Mistral$_{7B\text{-}inst}$ | 7.25B | 32 | 32 | 8 | 1024 |

alongside the context, which leads to lower ROUGE-L scores. On the other hand, the translation results of masked models are more concentrated on the task and context words, so the w/ $\mathcal{M}$ scores of Qwen2.5$_{14B\text{-}inst}$ increase significantly compared to the w/ instruction scores (Table 29). This implies that attention heads could influence the response style of the model. We provide a translation example in Table 36.

- For some small models, attention head mask $\mathcal{M}$ may fail on translations to relatively low-resource languages (e.g., translating to Arabic using models smaller than Qwen2.5$_{3B\text{-}inst}$). Theoretically, if a model has more attention heads, then it can form more functional pathways through their combinations. We suggest that this phenomenon of attention heads exhibiting functionality is better applicable to larger LLMs.

- We also test the attention head mask on Gemma 2 series (Team, 2024a). While on Gemma2$_{9B\text{-}inst}$ the masked model can also directly elicit the functionality, the masked model on Gemma2$_{2B\text{-}inst}$ fails on all the translation pairs. We would ascribe this to the number of attention heads in the model (208 heads in total, 26 layers and 8 heads for each). Although the model has more parameters than Qwen2.5$_{0.5B\text{-}inst}$ and Qwen2.5$_{0.5B\text{-}inst}$, the lack of sufficient granularity prevents it from acquiring a combination of attention heads that can directly elicit functionalities.

For the attention head masks in pre-trained models, we also provide full results (Table 37, 38) as complementary to the examples in Table 5. Specifically, we calculate the math expectation of Jaccard similarity on two randomly shuffled masks ("Random" column) as follows:

$$
\begin{aligned}
\mathbb{E}\left[\text{Jaccard}(\mathcal{M}_1, \mathcal{M}_2)\right] &= \frac{\mathbb{E}\left[|\mathcal{M}_1 \cap \mathcal{M}_2|\right]}{\mathbb{E}\left[|\mathcal{M}_1 \cup \mathcal{M}_2|\right]} \\
&= \frac{|\mathbf{1}| \frac{|\mathcal{M}_1|}{|\mathbf{1}|} \frac{|\mathcal{M}_2|}{|\mathbf{1}|}}{\mathbb{E}\left[|\mathcal{M}_1| + |\mathcal{M}_2| - |\mathcal{M}_1 \cap \mathcal{M}_2|\right]} \\
&= \frac{\frac{|\mathcal{M}_1||\mathcal{M}_2|}{|\mathbf{1}|}}{|\mathcal{M}_1| + |\mathcal{M}_2| - \frac{|\mathcal{M}_1||\mathcal{M}_2|}{|\mathbf{1}|}}
\end{aligned}
\tag{4}
$$

*Table 10.* Simple task datasets and their task instructions used in this paper.

| Task name | Samples | Prompt |
| --- | --- | --- |
| lowercase_first_letter | 814 | Output the first letter of the given word in lowercase.\n\nInput:\n\n{input}\n\nOutput:\n\n |
| park-country | 749 | Identify the country where the given national park is located.\n\nInput:\n\n{input}\n\nOutput:\n\n |
| synonym | 2880 | Identify a synonym for the given word.\n\nInput:\n\n{input}\n\nOutput:\n\n |
| ag_news | 7600 | Classify the given news headline into one of the categories: Business, Science, Sports, or World. Provide only the category name.\n\nInput:\n\n{input}\n\nOutput:\n\n |
| word_length | 814 | Determine the number of letters in the given word and output the count.\n\nInput:\n\n{input}\n\nOutput:\n\n |
| present-past | 293 | Convert the given verb from its present tense to its simple past tense.\n\nInput:\n\n{input}\n\nOutput:\n\n |
| capitalize | 813 | Output the given word with its first letter capitalized.\n\nInput:\n\n{input}\n\nOutput:\n\n |
| landmark-country | 836 | Identify the country where the given landmark is located.\n\nInput:\n\n{input}\n\nOutput:\n\n |
| english-german | 5145 | Translate the given English word into German.\n\nInput:\n\n{input}\n\nOutput:\n\n |
| sentiment | 1167 | Determine the sentiment of the given input. Output either 'positive' or 'negative'.\n\nInput:\n\n{input}\n\nOutput:\n\n |
| country-capital | 197 | What is the capital of the given country? Provide only the name of the capital.\n\nInput:\n\n{input}\n\nOutput:\n\n |
| person-occupation | 821 | Identify the occupation of the given individual.\n\nInput:\n\n{input}\n\nOutput:\n\n |
| country-currency | 197 | What is the official currency of the given country?\n\nInput:\n\n{input}\n\nOutput:\n\n |
| lowercase_last_letter | 814 | Output the last letter of the given word in lowercase.\n\nInput:\n\n{input}\n\nOutput:\n\n |
| person-sport | 318 | Identify the sport associated with the given individual.\n\nInput:\n\n{input}\n\nOutput:\n\n |
| person-instrument | 510 | Identify the musical instrument played by the given musician.\n\nInput:\n\n{input}\n\nOutput:\n\n |
| antonym | 2398 | Identify the antonym of the given word.\n\nInput:\n\n{input}\n\nOutput:\n\n |
| capitalize_last_letter | 814 | Output the last letter of the given word in uppercase.\n\nInput:\n\n{input}\n\nOutput:\n\n |
| english-french | 4698 | Translate the given English word into French.\n\nInput:\n\n{input}\n\nOutput:\n\n |
| next_item | 225 | What is the next sequential item following the given input?\n\nInput:\n\n{input}\n\nOutput:\n\n |
| singular-plural | 205 | Provide the plural form of the given singular noun.\n\nInput:\n\n{input}\n\nOutput:\n\n |
| capitalize_second_letter | 787 | Output the second letter of the given word in uppercase.\n\nInput:\n\n{input}\n\nOutput:\n\n |
| prev_item | 225 | What is the item that comes before the given input in a sequential context?\n\nInput:\n\n{input}\n\nOutput:\n\n |
| capitalize_first_letter | 813 | Output the first letter of the given word in uppercase.\n\nInput:\n\n{input}\n\nOutput:\n\n |
| english-spanish | 5199 | Translate the given English word into Spanish.\n\nInput:\n\n{input}\n\nOutput:\n\n |
| next_capital_letter | 814 | What is the next uppercase letter in alphabetical order after the given input?\n\nInput:\n\n{input}\n\nOutput:\n\n |
| national_parks | 451 | Identify the U.S. state where the given national park is located.\n\nInput:\n\n{input}\n\nOutput:\n\n |
| product-company | 522 | Identify the company associated with the given product.\n\nInput:\n\n{input}\n\nOutput:\n\n |
| conll2003_organization | 3843 | Extract the organization mentioned in the given text.\n\nInput:\n\n{input}\n\nOutput:\n\n |
| conll2003_person | 3544 | Extract the name of the person mentioned in the given text.\n\nInput:\n\n{input}\n\nOutput:\n\n |
| conll2003_location | 4499 | Extract the location mentioned in the given text.\n\nInput:\n\n{input}\n\nOutput:\n\n |
| adjective_v_verb_3 | 1000 | From the given words, identify the one that is an adjective.\n\nInput:\n\n{input}\n\nOutput:\n\n |
| object_v_concept_3 | 1000 | From the given words, identify the one that is a object.\n\nInput:\n\n{input}\n\nOutput:\n\n |
| verb_v_adjective_3 | 1000 | From the given words, identify the one that is a verb.\n\nInput:\n\n{input}\n\nOutput:\n\n |
| fruit_v_animal_3 | 1000 | From the given words, identify the one that is a fruit.\n\nInput:\n\n{input}\n\nOutput:\n\n |

*Table 11.* Hyper-parameters used in the mask training.

| Hyper-parameter | Translation task | Simple task |
| --- | --- | --- |
| Mask weight $M$ initialize | $\mathcal{N}(4, 0.02)$ | $\mathcal{N}(4, 0.02)$ |
| - Negative initialize in Table 6 | $\mathcal{N}(-3, 0.02)$ | $\mathcal{N}(-3, 0.02)$ |
| Batch size | 16 | 16 |
| Train epochs | 10 | – |
| Train steps | 6190 | 6250 |
| Learning rate | [1e-2, 1e-4], cosine scheduler | [1e-2, 1e-4], cosine scheduler |
| Warmup ratio | 0.1 | 0.1 |
| Gumbel temperature $\tau$ | [4, 0.05], linear scheduler | [4, 0.05], linear scheduler |
| - $\tau$ decrease ratio | 0.4 | 0.4 |

where $|\mathcal{M}|$ is the number of selected heads in the mask $\mathcal{M}$, and $|\mathbf{1}|$ (full mask) is the total number of heads in the model. As the selected heads in the instruction model are always more than the pre-trained model, we also calculate the recall rate on the masks of instruction models. These high recall rates across tasks indicate that the attention head masks can be largely inherited from the pre-trained models, therefore such functionalities probably have been already learned in the pre-training stage.

## C.2. Visualization of Attention Head Masks

We illustrate the attention head mask weight $\sigma(\mathbf{M})$ for all 42 translation pairs on Llama3.1$_{\text{8B-inst}}$ in Figure 7. From this we can draw some intuitive conclusions: (1) The removed attention heads do not exhibit a clear distribution pattern across the layers of the model. Heads used in the latter half of the model are only slightly more than those in the earlier layers. This indicates that the functionality of attention heads is widely distributed, with attention heads in different layers contributing to functional pathways. (2) When the model switches from instruction-following functionality to translation functionality, the removed functional components are unlikely to be located in the KV heads. This is because Llama3.1$_{\text{8B-inst}}$ features a grouped structure for attention heads (divided into 8 groups of 8 KV heads each, with each group containing 4 Q heads). If the original functionality rely on KV heads, the corresponding group of heads would consistently go unused. However, this pattern of all 4 attention heads in a row being unused is not evident in the figure. This suggests that such functionality is more likely distributed in the Q heads or O heads (the corresponding parts of the projection matrix $\mathbf{W}_O$).

*Table 12.* XNLI translation perplexity (PPL) on Llama3.1$_{8B-inst}$.

**Original**

| From \ To | en | zh | fr | es | de | ru | ar |
|---|---|---|---|---|---|---|---|
| en | — | 2.643 | 2.172 | 2.319 | 2.314 | 2.327 | 2.730 |
| zh | 2.143 | — | 2.093 | 2.339 | 2.257 | 2.304 | 2.925 |
| fr | 2.439 | 2.911 | — | 2.357 | 2.296 | 2.280 | 2.751 |
| es | 2.254 | 2.766 | 2.130 | — | 2.388 | 2.273 | 2.909 |
| de | 2.320 | 2.816 | 2.176 | 2.388 | — | 2.210 | 2.744 |
| ru | 2.621 | 3.532 | 2.240 | 2.513 | 2.377 | — | 3.085 |
| ar | 2.720 | 3.668 | 2.318 | 2.662 | 2.608 | 2.591 | — |

**w/ instruction**

| From \ To | en | zh | fr | es | de | ru | ar |
|---|---|---|---|---|---|---|---|
| en | — | 1.091 | 0.619 | 0.550 | 0.627 | 0.954 | 1.303 |
| zh | 1.003 | — | 0.945 | 0.988 | 1.109 | 1.233 | 1.645 |
| fr | 0.756 | 1.342 | — | 0.825 | 0.912 | 1.090 | 1.486 |
| es | 0.558 | 1.206 | 0.705 | — | 0.781 | 1.030 | 1.426 |
| de | 0.629 | 1.303 | 0.759 | 0.758 | — | 1.026 | 1.472 |
| ru | 1.096 | 1.555 | 0.973 | 1.040 | 1.073 | — | 1.701 |
| ar | 1.218 | 1.691 | 1.167 | 1.230 | 1.274 | 1.397 | — |

**w/ random $\mathcal{M}$**

| From \ To | en | zh | fr | es | de | ru | ar |
|---|---|---|---|---|---|---|---|
| en | — | 2.909 | 2.370 | 2.762 | 2.482 | 2.677 | 3.204 |
| zh | 2.250 | — | 2.208 | 2.745 | 2.257 | 2.791 | 3.320 |
| fr | 2.645 | 3.268 | — | 2.838 | 2.138 | 3.039 | 2.929 |
| es | 2.187 | 3.039 | 2.629 | — | 2.257 | 2.383 | 3.024 |
| de | 2.128 | 3.188 | 1.969 | 2.391 | — | 2.718 | 2.964 |
| ru | 2.655 | 2.791 | 2.198 | 2.568 | 3.223 | — | 4.732 |
| ar | 3.018 | 3.674 | 2.775 | 2.602 | 2.821 | 2.556 | — |

**w/ $\mathcal{M}$**

| From \ To | en | zh | fr | es | de | ru | ar |
|---|---|---|---|---|---|---|---|
| en | — | 1.017 | 0.552 | 0.504 | 0.580 | 1.055 | 1.322 |
| zh | 0.997 | — | 0.970 | 1.036 | 1.115 | 1.241 | 1.689 |
| fr | 0.735 | 1.314 | — | 0.805 | 0.909 | 1.098 | 1.504 |
| es | 0.551 | 1.177 | 0.699 | — | 0.805 | 1.076 | 1.487 |
| de | 0.639 | 1.242 | 0.758 | 0.765 | — | 1.050 | 1.501 |
| ru | 1.070 | 1.486 | 0.927 | 1.032 | 1.073 | — | 1.697 |
| ar | 1.175 | 1.657 | 1.131 | 1.211 | 1.286 | 1.464 | — |

**w/ $\mathcal{M}$ of Llama3.1$_{8B}$**

| From \ To | en | zh | fr | es | de | ru | ar |
|---|---|---|---|---|---|---|---|
| en | — | 1.017 | 0.552 | 0.504 | 0.580 | 1.055 | 1.322 |
| zh | 0.997 | — | 0.970 | 1.036 | 1.115 | 1.241 | 1.689 |
| fr | 0.735 | 1.314 | — | 0.805 | 0.909 | 1.098 | 1.504 |
| es | 0.551 | 1.177 | 0.699 | — | 0.805 | 1.076 | 1.487 |
| de | 0.639 | 1.242 | 0.758 | 0.765 | — | 1.050 | 1.501 |
| ru | 1.070 | 1.486 | 0.927 | 1.032 | 1.073 | — | 1.697 |
| ar | 1.175 | 1.657 | 1.131 | 1.211 | 1.286 | 1.464 | — |

*Table 13.* XNLI translation perplexity (PPL) on Llama3.1$_{8B}$.

**Original**

| From \ To | en | zh | fr | es | de | ru | ar |
|---|---|---|---|---|---|---|---|
| en | — | 2.308 | 1.672 | 1.853 | 1.780 | 1.824 | 2.434 |
| zh | 1.947 | — | 2.071 | 2.335 | 2.298 | 2.186 | 2.918 |
| fr | 1.714 | 2.573 | — | 2.002 | 1.951 | 1.932 | 2.516 |
| es | 1.543 | 2.446 | 1.704 | — | 1.904 | 1.863 | 2.465 |
| de | 1.592 | 2.532 | 1.778 | 2.008 | — | 1.854 | 2.574 |
| ru | 1.960 | 2.704 | 1.886 | 2.202 | 2.087 | — | 2.670 |
| ar | 2.201 | 2.914 | 2.053 | 2.312 | 2.340 | 2.282 | — |

**w/ instruction**

| From \ To | en | zh | fr | es | de | ru | ar |
|---|---|---|---|---|---|---|---|
| en | — | 1.901 | 1.240 | 1.350 | 1.258 | 1.573 | 1.945 |
| zh | 1.542 | — | 1.492 | 1.658 | 1.621 | 1.685 | 2.183 |
| fr | 1.523 | 2.136 | — | 1.543 | 1.525 | 1.660 | 2.086 |
| es | 1.327 | 1.955 | 1.289 | — | 1.402 | 1.609 | 2.029 |
| de | 1.410 | 2.080 | 1.366 | 1.499 | — | 1.614 | 2.107 |
| ru | 1.902 | 2.279 | 1.624 | 1.862 | 1.773 | — | 2.294 |
| ar | 1.867 | 2.410 | 1.754 | 1.942 | 1.908 | 1.961 | — |

**w/ random $\mathcal{M}$**

| From \ To | en | zh | fr | es | de | ru | ar |
|---|---|---|---|---|---|---|---|
| en | — | 2.703 | 2.329 | 2.337 | 2.133 | 2.377 | 3.251 |
| zh | 2.041 | — | 2.578 | 2.699 | 2.934 | 2.912 | 3.753 |
| fr | 1.727 | 2.875 | — | 3.190 | 2.407 | 2.187 | 2.916 |
| es | 2.221 | 2.630 | 1.849 | — | 2.841 | 2.115 | 3.179 |
| de | 2.511 | 3.247 | 2.055 | 2.445 | — | 2.190 | 4.044 |
| ru | 1.935 | 3.710 | 2.139 | 2.430 | 3.188 | — | 3.328 |
| ar | 2.567 | 3.440 | 2.596 | 2.958 | 2.680 | 4.758 | — |

**w/ $\mathcal{M}$**

| From \ To | en | zh | fr | es | de | ru | ar |
|---|---|---|---|---|---|---|---|
| en | — | 1.158 | 0.585 | 0.577 | 0.660 | 1.142 | 1.867 |
| zh | 1.087 | — | 1.111 | 1.142 | 1.300 | 1.349 | 1.960 |
| fr | 0.798 | 1.450 | — | 0.923 | 1.027 | 1.161 | 1.734 |
| es | 0.623 | 1.336 | 0.800 | — | 0.911 | 1.087 | 1.709 |
| de | 0.701 | 1.386 | 0.846 | 0.832 | — | 1.108 | 1.738 |
| ru | 1.124 | 1.641 | 1.037 | 1.137 | 1.172 | — | 1.998 |
| ar | 1.275 | 1.810 | 1.294 | 1.324 | 1.404 | 1.506 | — |

**w/ $\mathcal{M}$ of Llama3.1$_{8B-inst}$**

| From \ To | en | zh | fr | es | de | ru | ar |
|---|---|---|---|---|---|---|---|
| en | — | 2.213 | 1.512 | 1.515 | 1.480 | 1.754 | 2.393 |
| zh | 1.566 | — | 1.661 | 1.967 | 1.933 | 2.014 | 2.692 |
| fr | 1.312 | 2.051 | — | 1.742 | 1.589 | 1.596 | 2.363 |
| es | 1.048 | 1.915 | 1.410 | — | 1.539 | 1.759 | 2.395 |
| de | 1.224 | 1.998 | 1.566 | 1.730 | — | 1.658 | 2.446 |
| ru | 1.618 | 2.367 | 1.612 | 1.908 | 1.690 | — | 2.563 |
| ar | 1.785 | 2.334 | 1.809 | 1.901 | 1.868 | 1.939 | — |

*Table 14.* XNLI translation perplexity (PPL) on Qwen2.5$_{\text{7B-inst}}$.

**Original**

| From \ To | en | zh | fr | es | de | ru | ar |
|---|---|---|---|---|---|---|---|
| en | — | 5.481 | 3.268 | 3.466 | 3.171 | 3.109 | 3.925 |
| zh | 3.983 | — | 3.418 | 3.712 | 3.612 | 3.019 | 3.920 |
| fr | 4.178 | 5.962 | — | 3.682 | 3.443 | 2.884 | 3.964 |
| es | 3.799 | 5.809 | 3.183 | — | 3.112 | 2.754 | 4.092 |
| de | 3.673 | 5.627 | 3.328 | 3.722 | — | 2.820 | 4.065 |
| ru | 4.665 | 6.459 | 3.618 | 4.036 | 3.611 | — | 4.600 |
| ar | 4.299 | 5.744 | 3.239 | 3.663 | 3.543 | 2.891 | — |

**w/ instruction**

| From \ To | en | zh | fr | es | de | ru | ar |
|---|---|---|---|---|---|---|---|
| en | — | 2.045 | 1.023 | 0.942 | 0.980 | 1.429 | 1.902 |
| zh | 2.488 | — | 2.008 | 2.036 | 1.863 | 1.924 | 2.469 |
| fr | 2.120 | 2.875 | — | 1.770 | 1.773 | 1.826 | 2.334 |
| es | 1.690 | 2.504 | 1.446 | — | 1.450 | 1.701 | 2.257 |
| de | 1.843 | 2.879 | 1.595 | 1.639 | — | 1.781 | 2.372 |
| ru | 3.105 | 3.742 | 2.171 | 2.367 | 2.064 | — | 2.825 |
| ar | 2.824 | 3.514 | 2.103 | 2.300 | 2.211 | 2.144 | — |

**w/ random $\mathcal{M}$**

| From \ To | en | zh | fr | es | de | ru | ar |
|---|---|---|---|---|---|---|---|
| en | — | 4.604 | 3.523 | 6.213 | 4.080 | 13.644 | 8.878 |
| zh | 3.349 | — | 7.187 | 6.858 | 6.744 | 3.784 | 6.103 |
| fr | 6.117 | 8.348 | — | 3.928 | 8.762 | 5.616 | 5.521 |
| es | 3.976 | 5.231 | 6.837 | — | 3.276 | 5.282 | 3.813 |
| de | 5.392 | 5.698 | 4.055 | 3.774 | — | 3.488 | 9.176 |
| ru | 4.078 | 4.440 | 5.415 | 8.665 | 3.820 | — | 5.574 |
| ar | 7.215 | 5.548 | 9.866 | 6.808 | 5.344 | 3.336 | — |

**w/ $\mathcal{M}$**

| From \ To | en | zh | fr | es | de | ru | ar |
|---|---|---|---|---|---|---|---|
| en | — | 1.158 | 0.884 | 0.708 | 0.800 | 1.070 | 1.720 |
| zh | 1.058 | — | 1.355 | 1.330 | 1.406 | 1.357 | 2.075 |
| fr | 0.876 | 1.588 | — | 1.035 | 1.147 | 1.242 | 2.042 |
| es | 0.735 | 1.395 | 0.986 | — | 1.021 | 1.216 | 1.974 |
| de | 0.840 | 1.572 | 1.149 | 1.135 | — | 1.225 | 2.103 |
| ru | 1.236 | 1.832 | 1.377 | 1.462 | 1.456 | — | 2.376 |
| ar | 1.368 | 2.068 | 1.444 | 1.602 | 1.582 | 1.521 | — |

**w/ $\mathcal{M}$ of Qwen2.5$_{\text{7B}}$**

| From \ To | en | zh | fr | es | de | ru | ar |
|---|---|---|---|---|---|---|---|
| en | — | 3.024 | 2.444 | 2.476 | 2.339 | 2.419 | 3.226 |
| zh | 1.943 | — | 2.338 | 2.552 | 2.401 | 2.497 | 3.550 |
| fr | 1.553 | 2.935 | — | 2.369 | 2.350 | 2.353 | 3.415 |
| es | 1.521 | 2.796 | 2.101 | — | 1.775 | 2.216 | 3.397 |
| de | 1.732 | 2.604 | 2.052 | 2.252 | — | 2.055 | 3.298 |
| ru | 2.411 | 3.169 | 2.761 | 2.741 | 2.334 | — | 3.389 |
| ar | 1.968 | 3.136 | 2.063 | 2.319 | 2.676 | 2.434 | — |

*Table 15.* XNLI translation perplexity (PPL) on Qwen2.5$_{\text{7B}}$.

**Original**

| From \ To | en | zh | fr | es | de | ru | ar |
|---|---|---|---|---|---|---|---|
| en | — | 2.612 | 1.844 | 2.024 | 1.907 | 1.763 | 2.702 |
| zh | 1.855 | — | 2.010 | 2.220 | 2.131 | 1.985 | 2.706 |
| fr | 1.931 | 3.117 | — | 2.188 | 2.156 | 1.909 | 2.771 |
| es | 1.622 | 2.808 | 1.872 | — | 2.010 | 1.782 | 2.662 |
| de | 1.872 | 3.071 | 2.045 | 2.285 | — | 1.878 | 2.832 |
| ru | 2.323 | 3.401 | 2.212 | 2.469 | 2.275 | — | 2.970 |
| ar | 2.193 | 3.306 | 2.086 | 2.361 | 2.392 | 2.093 | — |

**w/ instruction**

| From \ To | en | zh | fr | es | de | ru | ar |
|---|---|---|---|---|---|---|---|
| en | — | 1.692 | 1.067 | 0.998 | 1.171 | 1.371 | 2.005 |
| zh | 1.297 | — | 1.286 | 1.345 | 1.454 | 1.497 | 2.102 |
| fr | 1.453 | 1.985 | — | 1.309 | 1.405 | 1.451 | 2.031 |
| es | 1.262 | 1.794 | 1.212 | — | 1.323 | 1.415 | 2.027 |
| de | 1.451 | 1.917 | 1.255 | 1.273 | — | 1.462 | 2.036 |
| ru | 1.843 | 2.228 | 1.554 | 1.574 | 1.658 | — | 2.212 |
| ar | 1.783 | 2.372 | 1.595 | 1.661 | 1.784 | 1.705 | — |

**w/ random $\mathcal{M}$**

| From \ To | en | zh | fr | es | de | ru | ar |
|---|---|---|---|---|---|---|---|
| en | — | 8.193 | 7.783 | 7.990 | 3.947 | 3.884 | 5.011 |
| zh | 4.700 | — | 5.472 | 7.655 | 4.886 | 4.930 | 13.43 |
| fr | 2.671 | 6.117 | — | 5.439 | 3.106 | 4.681 | 3.915 |
| es | 1.913 | 5.683 | 2.719 | — | 5.643 | 3.796 | 4.019 |
| de | 5.234 | 3.973 | 3.909 | 6.127 | — | 7.972 | 3.662 |
| ru | 6.286 | 10.887 | 7.899 | 5.810 | 4.157 | — | 6.908 |
| ar | 2.598 | 7.619 | 4.024 | 5.524 | 4.220 | 2.606 | — |

**w/ $\mathcal{M}$**

| From \ To | en | zh | fr | es | de | ru | ar |
|---|---|---|---|---|---|---|---|
| en | — | 2.590 | 1.712 | 1.730 | 1.792 | 1.788 | 2.592 |
| zh | 1.680 | — | 1.944 | 2.050 | 2.116 | 1.945 | 2.658 |
| fr | 1.819 | 2.761 | — | 1.897 | 1.982 | 1.837 | 2.714 |
| es | 1.538 | 2.460 | 1.615 | — | 1.886 | 1.705 | 2.656 |
| de | 1.799 | 2.712 | 1.862 | 2.103 | — | 1.790 | 2.751 |
| ru | 2.102 | 3.162 | 2.031 | 2.156 | 2.139 | — | 2.262 |
| ar | 1.354 | 2.167 | 1.483 | 1.501 | 1.758 | 1.517 | — |

**w/ $\mathcal{M}$ of Qwen2.5$_{\text{7B-inst}}$**

| From \ To | en | zh | fr | es | de | ru | ar |
|---|---|---|---|---|---|---|---|
| en | — | 2.628 | 1.697 | 1.799 | 1.773 | 1.689 | 2.582 |
| zh | 1.641 | — | 1.924 | 2.087 | 2.012 | 1.854 | 2.661 |
| fr | 1.699 | 2.972 | — | 1.997 | 2.030 | 1.867 | 2.742 |
| es | 1.510 | 2.631 | 1.748 | — | 1.945 | 1.768 | 2.690 |
| de | 1.743 | 2.883 | 1.946 | 1.956 | — | 1.825 | 2.786 |
| ru | 2.110 | 3.271 | 2.187 | 2.409 | 2.182 | — | 2.991 |
| ar | 2.050 | 3.054 | 2.077 | 2.288 | 2.410 | 2.037 | — |

*Table 16.* XNLI translation perplexity (PPL) on Qwen2.5$_{\text{0.5B-inst}}$.

| Original | | | | | | | |
| To \ From | en | zh | fr | es | de | ru | ar |
|---|---|---|---|---|---|---|---|
| en | – | 3.045 | 2.100 | 2.386 | 2.545 | 2.254 | 3.365 |
| zh | 2.530 | – | 2.748 | 3.073 | 3.255 | 2.666 | 3.761 |
| fr | 2.608 | 4.223 | – | 3.056 | 3.224 | 2.608 | 4.020 |
| es | 2.505 | 4.117 | 2.646 | – | 3.360 | 2.579 | 3.955 |
| de | 2.727 | 4.306 | 2.804 | 3.361 | – | 2.723 | 4.228 |
| ru | 2.967 | 4.447 | 2.709 | 3.148 | 3.119 | – | 4.582 |
| ar | 3.511 | 4.919 | 3.066 | 3.493 | 3.771 | 3.302 | – |

| w/ instruction | | | | | | | |
| To \ From | en | zh | fr | es | de | ru | ar |
|---|---|---|---|---|---|---|---|
| en | – | 1.763 | 1.021 | 0.989 | 1.361 | 1.485 | 2.401 |
| zh | 1.601 | – | 1.574 | 1.688 | 2.061 | 1.864 | 2.726 |
| fr | 1.233 | 2.325 | – | 1.369 | 1.781 | 1.676 | 2.596 |
| es | 1.142 | 2.152 | 1.223 | – | 1.707 | 1.640 | 2.509 |
| de | 1.357 | 2.565 | 1.478 | 1.586 | – | 1.751 | 2.681 |
| ru | 1.911 | 2.784 | 1.746 | 1.858 | 2.077 | – | 2.711 |
| ar | 2.268 | 3.067 | 2.152 | 2.364 | 2.677 | 2.193 | – |

| w/ $\mathcal{M}$ | | | | | | | |
| To \ From | en | zh | fr | es | de | ru | ar |
|---|---|---|---|---|---|---|---|
| en | – | 1.939 | 1.427 | 2.070 | 1.874 | 1.707 | 3.128 |
| zh | 1.601 | – | 2.084 | 2.324 | 2.543 | 2.121 | 3.091 |
| fr | 1.210 | 2.358 | – | 2.047 | 2.170 | 1.954 | 3.038 |
| es | 1.113 | 2.528 | 1.870 | – | 2.263 | 2.569 | 3.185 |
| de | 1.328 | 2.483 | 2.111 | 2.322 | – | 2.030 | 3.090 |
| ru | 1.847 | 2.736 | 2.534 | 2.530 | 2.329 | – | 3.271 |
| ar | 4.174 | 3.016 | 2.540 | 2.638 | 2.874 | 2.547 | – |

*Table 18.* XNLI translation perplexity (PPL) on Qwen2.5$_{\text{3B-inst}}$.

| Original | | | | | | | |
| To \ From | en | zh | fr | es | de | ru | ar |
|---|---|---|---|---|---|---|---|
| en | – | 5.220 | 2.913 | 3.295 | 3.232 | 3.233 | 4.116 |
| zh | 4.120 | – | 3.383 | 3.883 | 3.726 | 3.290 | 4.425 |
| fr | 4.318 | 6.318 | – | 4.034 | 3.663 | 3.449 | 4.609 |
| es | 3.602 | 5.724 | 3.351 | – | 3.539 | 3.314 | 4.506 |
| de | 3.369 | 5.673 | 3.153 | 3.759 | – | 3.268 | 4.308 |
| ru | 4.658 | 6.772 | 3.570 | 4.193 | 3.883 | – | 5.203 |
| ar | 4.576 | 6.573 | 3.620 | 4.081 | 4.043 | 3.656 | – |

| w/ instruction | | | | | | | |
| To \ From | en | zh | fr | es | de | ru | ar |
|---|---|---|---|---|---|---|---|
| en | – | 1.864 | 0.919 | 0.828 | 1.008 | 1.277 | 1.930 |
| zh | 2.215 | – | 1.647 | 1.682 | 1.781 | 1.827 | 2.516 |
| fr | 1.781 | 2.811 | – | 1.476 | 1.604 | 1.615 | 2.267 |
| es | 1.390 | 2.360 | 1.267 | – | 1.368 | 1.539 | 2.163 |
| de | 1.607 | 2.748 | 1.370 | 1.347 | – | 1.651 | 2.289 |
| ru | 2.600 | 3.396 | 1.753 | 1.925 | 1.828 | – | 2.577 |
| ar | 2.737 | 3.620 | 2.035 | 2.110 | 2.207 | 2.105 | – |

| w/ $\mathcal{M}$ | | | | | | | |
| To \ From | en | zh | fr | es | de | ru | ar |
|---|---|---|---|---|---|---|---|
| en | – | 1.176 | 0.730 | 0.736 | 1.033 | 1.084 | 1.584 |
| zh | 1.141 | – | 1.318 | 1.336 | 1.823 | 1.586 | 2.380 |
| fr | 0.833 | 1.615 | – | 1.231 | 1.463 | 1.313 | 1.902 |
| es | 0.729 | 1.489 | 0.901 | – | 1.345 | 1.299 | 1.892 |
| de | 0.840 | 1.604 | 1.102 | 1.167 | – | 1.236 | 1.896 |
| ru | 1.261 | 1.955 | 1.198 | 1.356 | 1.508 | – | 2.056 |
| ar | 1.457 | 2.121 | 1.528 | 1.748 | 1.997 | 1.739 | – |

*Table 17.* XNLI translation perplexity (PPL) on Qwen2.5$_{\text{1.5B-inst}}$.

| Original | | | | | | | |
| To \ From | en | zh | fr | es | de | ru | ar |
|---|---|---|---|---|---|---|---|
| en | – | 2.941 | 1.919 | 2.025 | 2.138 | 1.984 | 2.918 |
| zh | 2.327 | – | 2.475 | 2.716 | 2.765 | 2.421 | 3.438 |
| fr | 2.679 | 4.329 | – | 2.987 | 2.873 | 2.457 | 3.709 |
| es | 2.368 | 4.306 | 2.518 | – | 2.903 | 2.441 | 3.853 |
| de | 2.385 | 4.022 | 2.572 | 2.935 | – | 2.442 | 3.584 |
| ru | 3.023 | 4.627 | 2.606 | 3.109 | 2.833 | – | 4.093 |
| ar | 3.142 | 4.450 | 2.718 | 3.016 | 3.287 | 3.023 | – |

| w/ instruction | | | | | | | |
| To \ From | en | zh | fr | es | de | ru | ar |
|---|---|---|---|---|---|---|---|
| en | – | 1.545 | 0.867 | 0.835 | 0.984 | 1.198 | 1.886 |
| zh | 1.429 | – | 1.433 | 1.440 | 1.628 | 1.515 | 2.238 |
| fr | 1.303 | 2.153 | – | 1.300 | 1.476 | 1.472 | 2.108 |
| es | 1.052 | 1.879 | 1.081 | – | 1.324 | 1.340 | 2.024 |
| de | 1.153 | 2.086 | 1.237 | 1.252 | – | 1.413 | 2.181 |
| ru | 1.794 | 2.551 | 1.497 | 1.623 | 1.652 | – | 2.303 |
| ar | 2.019 | 2.718 | 1.850 | 2.015 | 2.170 | 1.864 | – |

| w/ $\mathcal{M}$ | | | | | | | |
| To \ From | en | zh | fr | es | de | ru | ar |
|---|---|---|---|---|---|---|---|
| en | – | 1.354 | 0.889 | 0.854 | 1.086 | 1.177 | 2.018 |
| zh | 1.219 | – | 1.400 | 1.616 | 1.790 | 1.519 | 2.480 |
| fr | 0.996 | 2.050 | – | 1.411 | 1.516 | 1.319 | 2.236 |
| es | 0.802 | 2.109 | 1.041 | – | 1.411 | 1.292 | 2.134 |
| de | 0.901 | 1.872 | 1.332 | 1.442 | – | 1.363 | 2.290 |
| ru | 1.763 | 2.364 | 1.701 | 1.716 | 1.668 | – | 2.426 |
| ar | 1.637 | 2.472 | 1.694 | 1.920 | 2.080 | 1.868 | – |

*Table 19.* XNLI translation perplexity (PPL) on Qwen2.5$_{\text{14B-inst}}$.

| Original | | | | | | | |
| To \ From | en | zh | fr | es | de | ru | ar |
|---|---|---|---|---|---|---|---|
| en | – | 7.331 | 4.954 | 5.043 | 4.740 | 4.621 | 5.106 |
| zh | 6.797 | – | 5.683 | 6.016 | 5.718 | 5.021 | 5.413 |
| fr | 6.748 | 8.296 | – | 5.653 | 5.345 | 4.956 | 5.438 |
| es | 6.342 | 7.958 | 5.225 | – | 5.167 | 4.893 | 5.328 |
| de | 5.895 | 7.858 | 4.887 | 5.235 | – | 4.630 | 5.148 |
| ru | 7.253 | 8.558 | 5.587 | 6.065 | 5.533 | – | 5.539 |
| ar | 6.358 | 7.803 | 5.092 | 5.597 | 5.272 | 4.625 | – |

| w/ instruction | | | | | | | |
| To \ From | en | zh | fr | es | de | ru | ar |
|---|---|---|---|---|---|---|---|
| en | – | 3.372 | 1.732 | 1.485 | 1.553 | 2.015 | 2.467 |
| zh | 3.805 | – | 3.081 | 3.116 | 3.046 | 2.919 | 3.707 |
| fr | 3.887 | 5.032 | – | 2.736 | 2.644 | 2.699 | 3.296 |
| es | 2.915 | 4.250 | 2.253 | – | 2.192 | 2.355 | 3.090 |
| de | 3.103 | 4.749 | 2.408 | 2.340 | – | 2.396 | 3.009 |
| ru | 5.059 | 6.196 | 3.166 | 3.504 | 3.081 | – | 3.794 |
| ar | 4.787 | 5.923 | 3.316 | 3.627 | 3.227 | 2.963 | – |

| w/ $\mathcal{M}$ | | | | | | | |
| To \ From | en | zh | fr | es | de | ru | ar |
|---|---|---|---|---|---|---|---|
| en | – | 1.000 | 0.524 | 0.440 | 0.564 | 0.739 | 1.195 |
| zh | 0.937 | – | 0.877 | 0.930 | 1.052 | 1.003 | 1.542 |
| fr | 0.791 | 1.309 | – | 0.781 | 0.896 | 0.902 | 1.426 |
| es | 0.590 | 1.186 | 0.651 | – | 0.788 | 0.828 | 1.335 |
| de | 0.665 | 1.313 | 0.746 | 0.744 | – | 0.877 | 1.397 |
| ru | 1.078 | 1.550 | 0.917 | 0.978 | 1.047 | – | 1.561 |
| ar | 1.131 | 1.648 | 1.026 | 1.115 | 1.186 | 1.108 | – |

*Table 20.* XNLI translation perplexity (PPL) on Phi3.5$_{\text{mini-inst}}$.

**Original**

| From \ To | en | zh | fr | es | de | ru | ar |
|---|---|---|---|---|---|---|---|
| en | – | 3.337 | 3.262 | 3.682 | 3.773 | 3.698 | 2.126 |
| zh | 6.035 | – | 4.509 | 5.159 | 5.003 | 5.202 | 2.884 |
| fr | 5.627 | 4.046 | – | 4.893 | 5.054 | 4.606 | 2.479 |
| es | 5.673 | 3.992 | 4.392 | – | 4.918 | 4.669 | 2.665 |
| de | 5.276 | 3.978 | 4.378 | 4.961 | – | 4.487 | 2.504 |
| ru | 6.271 | 4.341 | 4.427 | 5.168 | 4.883 | – | 2.756 |
| ar | 6.192 | 4.419 | 4.440 | 4.915 | 5.367 | 4.965 | – |

**w/ instruction**

| From \ To | en | zh | fr | es | de | ru | ar |
|---|---|---|---|---|---|---|---|
| en | – | 2.256 | 2.710 | 2.752 | 2.965 | 2.695 | 1.665 |
| zh | 4.958 | – | 4.024 | 4.396 | 4.382 | 3.563 | 2.145 |
| fr | 3.893 | 2.808 | – | 3.810 | 3.919 | 3.208 | 1.963 |
| es | 3.602 | 2.707 | 3.457 | – | 3.704 | 3.184 | 1.926 |
| de | 3.713 | 2.770 | 3.543 | 3.704 | – | 3.142 | 2.005 |
| ru | 5.315 | 3.126 | 4.286 | 4.575 | 4.511 | – | 2.222 |
| ar | 5.165 | 3.359 | 4.283 | 4.601 | 4.692 | 3.749 | – |

**w/ $\mathcal{M}$**

| From \ To | en | zh | fr | es | de | ru | ar |
|---|---|---|---|---|---|---|---|
| en | – | 0.945 | 0.589 | 0.549 | 0.722 | 1.198 | 1.050 |
| zh | 1.263 | – | 1.185 | 1.320 | 1.381 | 1.892 | 1.198 |
| fr | 0.813 | 1.153 | – | 0.900 | 1.129 | 1.603 | 1.045 |
| es | 0.664 | 1.080 | 0.808 | – | 1.016 | 1.492 | 0.973 |
| de | 0.739 | 1.131 | 0.868 | 0.946 | – | 1.404 | 1.161 |
| ru | 1.320 | 1.349 | 1.179 | 1.271 | 1.368 | – | 1.184 |
| ar | 1.463 | 1.523 | 1.384 | 1.484 | 1.728 | 8.590 | – |

*Table 21.* XNLI translation perplexity (PPL) on Mistral$_{\text{7B-inst}}$.

**Original**

| From \ To | en | zh | fr | es | de | ru | ar |
|---|---|---|---|---|---|---|---|
| en | – | 3.667 | 3.032 | 3.264 | 3.333 | 2.821 | 2.381 |
| zh | 4.457 | – | 3.174 | 3.261 | 3.512 | 3.014 | 2.184 |
| fr | 3.438 | 3.118 | – | 2.863 | 3.032 | 2.517 | 2.163 |
| es | 3.579 | 3.190 | 2.787 | – | 3.026 | 2.570 | 2.140 |
| de | 3.460 | 3.329 | 2.650 | 2.791 | – | 2.483 | 2.265 |
| ru | 4.512 | 3.551 | 2.931 | 3.186 | 3.204 | – | 2.207 |
| ar | 4.304 | 3.551 | 3.077 | 3.315 | 3.527 | 2.981 | – |

**w/ instruction**

| From \ To | en | zh | fr | es | de | ru | ar |
|---|---|---|---|---|---|---|---|
| en | – | 1.910 | 1.327 | 1.267 | 1.516 | 1.560 | 1.600 |
| zh | 2.103 | – | 1.928 | 2.052 | 2.238 | 2.022 | 1.834 |
| fr | 1.394 | 1.947 | – | 1.410 | 1.603 | 1.509 | 1.653 |
| es | 1.147 | 1.915 | 1.342 | – | 1.467 | 1.553 | 1.628 |
| de | 1.215 | 1.937 | 1.320 | 1.271 | – | 1.513 | 1.670 |
| ru | 1.979 | 2.288 | 1.636 | 1.765 | 1.892 | – | 1.713 |
| ar | 2.409 | 2.879 | 2.255 | 2.404 | 2.469 | 2.262 | – |

**w/ $\mathcal{M}$**

| From \ To | en | zh | fr | es | de | ru | ar |
|---|---|---|---|---|---|---|---|
| en | – | 3.275 | 2.440 | 3.540 | 2.524 | 2.854 | 2.654 |
| zh | 2.914 | – | 3.605 | 3.063 | 3.275 | 3.170 | 2.540 |
| fr | 3.518 | 3.338 | – | 2.248 | 3.562 | 2.744 | 2.061 |
| es | 2.377 | 3.159 | 4.525 | – | 3.784 | 2.572 | 2.358 |
| de | 2.474 | 4.737 | 2.720 | 3.608 | – | 2.944 | 2.828 |
| ru | 3.547 | 4.554 | 3.442 | 3.492 | 4.710 | – | 2.916 |
| ar | 3.349 | 4.378 | 5.172 | 3.782 | 3.962 | 3.510 | – |

*Table 22.* XNLI translation ROUGE-L on Llama3.1$_{\text{8B-inst}}$.

**Original**

| From \ To | en | zh | fr | es | de | ru | ar |
|---|---|---|---|---|---|---|---|
| en | – | 0.020 | 0.082 | 0.069 | 0.069 | 0.045 | 0.029 |
| zh | 0.021 | – | 0.019 | 0.024 | 0.023 | 0.025 | 0.013 |

**w/ instruction**

| From \ To | en | zh | fr | es | de | ru | ar |
|---|---|---|---|---|---|---|---|
| en | – | 0.539 | 0.541 | 0.599 | 0.576 | 0.491 | 0.414 |
| zh | 0.669 | – | 0.519 | 0.556 | 0.501 | 0.423 | 0.352 |

**w/ $\mathcal{M}$**

| From \ To | en | zh | fr | es | de | ru | ar |
|---|---|---|---|---|---|---|---|
| en | – | 0.589 | 0.711 | 0.748 | 0.677 | 0.401 | 0.427 |
| zh | 0.656 | – | 0.474 | 0.518 | 0.488 | 0.384 | 0.334 |

**w/ $\mathcal{M}$ of Llama3.1$_{\text{8B}}$**

| From \ To | en | zh | fr | es | de | ru | ar |
|---|---|---|---|---|---|---|---|
| en | – | 0.019 | 0.204 | 0.241 | 0.099 | 0.046 | 0.045 |
| zh | 0.344 | – | 0.442 | 0.068 | 0.227 | 0.071 | 0.039 |

*Table 23.* XNLI translation ROUGE-L on Llama3.1$_{\text{8B}}$.

**Original**

| From \ To | en | zh | fr | es | de | ru | ar |
|---|---|---|---|---|---|---|---|
| en | – | 0.017 | 0.072 | 0.067 | 0.066 | 0.045 | 0.030 |
| zh | 0.020 | – | 0.017 | 0.019 | 0.022 | 0.023 | 0.010 |

**w/ instruction**

| From \ To | en | zh | fr | es | de | ru | ar |
|---|---|---|---|---|---|---|---|
| en | – | 0.017 | 0.084 | 0.080 | 0.076 | 0.054 | 0.029 |
| zh | 0.140 | – | 0.078 | 0.080 | 0.071 | 0.059 | 0.034 |

**w/ $\mathcal{M}$**

| From \ To | en | zh | fr | es | de | ru | ar |
|---|---|---|---|---|---|---|---|
| en | – | 0.424 | 0.658 | 0.657 | 0.606 | 0.179 | 0.067 |
| zh | 0.581 | – | 0.406 | 0.443 | 0.311 | 0.258 | 0.129 |

**w/ $\mathcal{M}$ of Llama3.1$_{\text{8B-inst}}$**

| From \ To | en | zh | fr | es | de | ru | ar |
|---|---|---|---|---|---|---|---|
| en | – | 0.016 | 0.071 | 0.074 | 0.076 | 0.037 | 0.030 |
| zh | 0.134 | – | 0.044 | 0.038 | 0.043 | 0.023 | 0.012 |

*Table 24.* XNLI translation ROUGE-L on Qwen2.5$_{7B\text{-inst}}$.

| Original | | | | | | |
| --- | --- | --- | --- | --- | --- | --- |
| From \ To | en | zh | fr | es | de | ru | ar |
| en | – | 0.017 | 0.076 | 0.067 | 0.066 | 0.043 | 0.027 |
| zh | 0.014 | – | 0.011 | 0.013 | 0.019 | 0.019 | 0.006 |

| w/ instruction | | | | | | |
| --- | --- | --- | --- | --- | --- | --- |
| From \ To | en | zh | fr | es | de | ru | ar |
| en | – | 0.570 | 0.581 | 0.635 | 0.580 | 0.405 | 0.387 |
| zh | 0.586 | – | 0.462 | 0.484 | 0.432 | 0.352 | 0.290 |

| w/ $\mathcal{M}$ | | | | | | |
| --- | --- | --- | --- | --- | --- | --- |
| From \ To | en | zh | fr | es | de | ru | ar |
| en | – | 0.599 | 0.390 | 0.583 | 0.522 | 0.253 | 0.142 |
| zh | 0.634 | – | 0.215 | 0.326 | 0.328 | 0.181 | 0.149 |

| w/ $\mathcal{M}$ of Qwen2.5$_{7B}$ | | | | | | |
| --- | --- | --- | --- | --- | --- | --- |
| From \ To | en | zh | fr | es | de | ru | ar |
| en | – | 0.020 | 0.079 | 0.066 | 0.057 | 0.042 | 0.025 |
| zh | 0.095 | – | 0.024 | 0.035 | 0.031 | 0.022 | 0.006 |

*Table 25.* XNLI translation ROUGE-L on Qwen2.5$_{7B}$.

| Original | | | | | | |
| --- | --- | --- | --- | --- | --- | --- |
| From \ To | en | zh | fr | es | de | ru | ar |
| en | – | 0.012 | 0.081 | 0.075 | 0.068 | 0.045 | 0.028 |
| zh | 0.019 | – | 0.014 | 0.014 | 0.021 | 0.019 | 0.009 |

| w/ instruction | | | | | | |
| --- | --- | --- | --- | --- | --- | --- |
| From \ To | en | zh | fr | es | de | ru | ar |
| en | – | 0.242 | 0.327 | 0.339 | 0.269 | 0.240 | 0.175 |
| zh | 0.371 | – | 0.237 | 0.239 | 0.217 | 0.205 | 0.166 |

| w/ $\mathcal{M}$ | | | | | | |
| --- | --- | --- | --- | --- | --- | --- |
| From \ To | en | zh | fr | es | de | ru | ar |
| en | – | 0.013 | 0.083 | 0.077 | 0.073 | 0.036 | 0.035 |
| zh | 0.127 | – | 0.035 | 0.042 | 0.047 | 0.032 | 0.008 |

| w/ $\mathcal{M}$ of Qwen2.5$_{7B\text{-inst}}$ | | | | | | |
| --- | --- | --- | --- | --- | --- | --- |
| From \ To | en | zh | fr | es | de | ru | ar |
| en | – | 0.016 | 0.095 | 0.066 | 0.061 | 0.052 | 0.024 |
| zh | 0.142 | – | 0.023 | 0.030 | 0.038 | 0.032 | 0.015 |

*Table 26.* XNLI translation ROUGE-L on Qwen2.5$_{0.5B\text{-inst}}$.

| Original | | | | | | |
| --- | --- | --- | --- | --- | --- | --- |
| From \ To | en | zh | fr | es | de | ru | ar |
| en | – | 0.015 | 0.083 | 0.066 | 0.069 | 0.052 | 0.031 |
| zh | 0.013 | – | 0.008 | 0.011 | 0.017 | 0.018 | 0.010 |

| w/ instruction | | | | | | |
| --- | --- | --- | --- | --- | --- | --- |
| From \ To | en | zh | fr | es | de | ru | ar |
| en | – | 0.505 | 0.499 | 0.498 | 0.456 | 0.264 | 0.242 |
| zh | 0.481 | – | 0.389 | 0.350 | 0.308 | 0.233 | 0.214 |

| w/ $\mathcal{M}$ | | | | | | |
| --- | --- | --- | --- | --- | --- | --- |
| From \ To | en | zh | fr | es | de | ru | ar |
| en | – | 0.367 | 0.296 | 0.081 | 0.296 | 0.133 | 0.044 |
| zh | 0.446 | – | 0.129 | 0.120 | 0.175 | 0.086 | 0.088 |

*Table 27.* XNLI translation ROUGE-L on Qwen2.5$_{1.5B\text{-inst}}$.

| Original | | | | | | |
| --- | --- | --- | --- | --- | --- | --- |
| From \ To | en | zh | fr | es | de | ru | ar |
| en | – | 0.014 | 0.077 | 0.060 | 0.067 | 0.043 | 0.029 |
| zh | 0.019 | – | 0.013 | 0.014 | 0.018 | 0.022 | 0.009 |

| w/ instruction | | | | | | |
| --- | --- | --- | --- | --- | --- | --- |
| From \ To | en | zh | fr | es | de | ru | ar |
| en | – | 0.557 | 0.543 | 0.570 | 0.544 | 0.343 | 0.279 |
| zh | 0.569 | – | 0.396 | 0.369 | 0.347 | 0.296 | 0.236 |

| w/ $\mathcal{M}$ | | | | | | |
| --- | --- | --- | --- | --- | --- | --- |
| From \ To | en | zh | fr | es | de | ru | ar |
| en | – | 0.505 | 0.522 | 0.518 | 0.461 | 0.211 | 0.152 |
| zh | 0.587 | – | 0.351 | 0.279 | 0.285 | 0.217 | 0.072 |

*Table 28.* XNLI translation ROUGE-L on Qwen2.5$_{3B\text{-inst}}$.

| Original | | | | | | |
| --- | --- | --- | --- | --- | --- | --- |
| From \ To | en | zh | fr | es | de | ru | ar |
| en | – | 0.016 | 0.082 | 0.068 | 0.070 | 0.045 | 0.029 |
| zh | 0.016 | – | 0.013 | 0.015 | 0.020 | 0.018 | 0.005 |

| w/ instruction | | | | | | |
| --- | --- | --- | --- | --- | --- | --- |
| From \ To | en | zh | fr | es | de | ru | ar |
| en | – | 0.581 | 0.618 | 0.625 | 0.527 | 0.356 | 0.305 |
| zh | 0.464 | – | 0.426 | 0.438 | 0.399 | 0.273 | 0.230 |

| w/ $\mathcal{M}$ | | | | | | |
| --- | --- | --- | --- | --- | --- | --- |
| From \ To | en | zh | fr | es | de | ru | ar |
| en | – | 0.615 | 0.567 | 0.574 | 0.416 | 0.243 | 0.352 |
| zh | 0.566 | – | 0.299 | 0.323 | 0.136 | 0.061 | 0.056 |

*Table 29.* XNLI translation ROUGE-L on Qwen2.5$_{\text{14B-inst}}$.

**Original**

| From \ To | en | zh | fr | es | de | ru | ar |
|---|---|---|---|---|---|---|---|
| en | – | 0.017 | 0.077 | 0.064 | 0.067 | 0.042 | 0.027 |
| zh | 0.013 | – | 0.010 | 0.012 | 0.018 | 0.017 | 0.004 |

**w/ instruction**

| From \ To | en | zh | fr | es | de | ru | ar |
|---|---|---|---|---|---|---|---|
| en | – | 0.498 | 0.599 | 0.588 | 0.506 | 0.416 | 0.247 |
| zh | 0.521 | – | 0.403 | 0.380 | 0.362 | 0.300 | 0.184 |

**w/ $\mathcal{M}$**

| From \ To | en | zh | fr | es | de | ru | ar |
|---|---|---|---|---|---|---|---|
| en | – | 0.676 | 0.716 | 0.761 | 0.660 | 0.534 | 0.454 |
| zh | 0.684 | – | 0.542 | 0.586 | 0.499 | 0.362 | 0.357 |

*Table 30.* XNLI translation ROUGE-L on Phi3.5$_{\text{mini-inst}}$.

**Original**

| From \ To | en | zh | fr | es | de | ru | ar |
|---|---|---|---|---|---|---|---|
| en | – | 0.018 | 0.086 | 0.076 | 0.073 | 0.048 | 0.032 |
| zh | 0.012 | – | 0.012 | 0.012 | 0.022 | 0.025 | 0.010 |

**w/ instruction**

| From \ To | en | zh | fr | es | de | ru | ar |
|---|---|---|---|---|---|---|---|
| en | – | 0.276 | 0.351 | 0.342 | 0.334 | 0.215 | 0.279 |
| zh | 0.234 | – | 0.228 | 0.216 | 0.208 | 0.152 | 0.183 |

**w/ $\mathcal{M}$**

| From \ To | en | zh | fr | es | de | ru | ar |
|---|---|---|---|---|---|---|---|
| en | – | 0.463 | 0.669 | 0.699 | 0.656 | 0.293 | 0.074 |
| zh | 0.525 | – | 0.406 | 0.399 | 0.400 | 0.132 | 0.156 |

*Table 31.* XNLI translation ROUGE-L on Mistral$_{\text{7B-inst}}$.

**Original**

| From \ To | en | zh | fr | es | de | ru | ar |
|---|---|---|---|---|---|---|---|
| en | – | 0.020 | 0.087 | 0.073 | 0.075 | 0.048 | 0.030 |
| zh | 0.022 | – | 0.020 | 0.019 | 0.028 | 0.029 | 0.011 |

**w/ instruction**

| From \ To | en | zh | fr | es | de | ru | ar |
|---|---|---|---|---|---|---|---|
| en | – | 0.264 | 0.431 | 0.433 | 0.378 | 0.282 | 0.177 |
| zh | 0.405 | – | 0.290 | 0.306 | 0.267 | 0.232 | 0.156 |

**w/ $\mathcal{M}$**

| From \ To | en | zh | fr | es | de | ru | ar |
|---|---|---|---|---|---|---|---|
| en | – | 0.547 | 0.663 | 0.656 | 0.673 | 0.461 | 0.199 |
| zh | 0.557 | – | 0.378 | 0.502 | 0.399 | 0.330 | 0.177 |

*Table 32.* Simple task accuracy on Llama3.1$_{\text{8B-inst}}$.

| Task name | Original | w/ instruction | w/ $\mathcal{M}$ | w/ $\mathcal{M}$ of Llama3.1$_{\text{8B}}$ |
|---|---|---|---|---|
| lowercase_first_letter | 0.00 | 0.61 | 1.00 | 0.99 |
| park-country | 0.18 | 0.65 | 0.68 | 0.67 |
| synonym | 0.01 | 0.29 | 0.48 | 0.51 |
| ag_news | 0.00 | 0.76 | 0.81 | 0.87 |
| word_length | 0.00 | 0.97 | 0.29 | 0.07 |
| present-past | 0.02 | 0.76 | 0.91 | 0.97 |
| capitalize | 0.10 | 1.00 | 0.98 | 0.97 |
| landmark-country | 0.03 | 0.86 | 0.87 | 0.87 |
| english-german | 0.00 | 0.43 | 0.61 | 0.47 |
| sentiment | 0.01 | 0.59 | 0.86 | 0.64 |
| country-capital | 0.04 | 0.92 | 0.85 | 0.80 |
| person-occupation | 0.17 | 0.19 | 0.68 | 0.53 |
| country-currency | 0.00 | 0.07 | 0.44 | 0.56 |
| lowercase_last_letter | 0.00 | 0.00 | 0.88 | 0.65 |
| person-sport | 0.38 | 0.01 | 0.91 | 0.86 |
| person-instrument | 0.06 | 0.05 | 0.80 | 0.77 |
| antonym | 0.12 | 0.30 | 0.76 | 0.75 |
| capitalize_last_letter | 0.01 | 0.14 | 0.96 | 0.08 |
| english-french | 0.01 | 0.60 | 0.77 | 0.70 |
| next_item | 0.01 | 0.06 | 0.78 | 0.64 |
| commonsense_qa | 0.00 | 0.33 | 0.70 | 0.17 |
| singular-plural | 0.02 | 0.95 | 0.97 | 0.99 |
| capitalize_second_letter | 0.07 | 0.00 | 0.63 | 0.17 |
| prev_item | 0.02 | 0.20 | 0.87 | 0.65 |
| capitalize_first_letter | 0.15 | 0.97 | 1.00 | 0.98 |
| english-spanish | 0.00 | 0.59 | 0.77 | 0.86 |
| next_capital_letter | 0.03 | 0.12 | 0.99 | 0.19 |
| national_parks | 0.04 | 0.65 | 0.64 | 0.63 |
| product-company | 0.06 | 0.73 | 0.83 | 0.78 |
| conll2003_organization | 0.04 | 0.13 | 0.57 | 0.37 |
| conll2003_person | 0.02 | 0.40 | 0.89 | 0.84 |
| conll2003_location | 0.00 | 0.41 | 0.92 | 0.75 |
| adjective_v_verb_3 | 0.00 | 0.14 | 0.99 | 0.97 |
| object_v_concept_3 | 0.00 | 0.02 | 0.98 | 0.99 |
| verb_v_adjective_3 | 0.00 | 0.23 | 0.98 | 0.96 |
| fruit_v_animal_3 | 0.00 | 0.51 | 0.98 | 0.87 |
| **Average** | 0.05 | 0.44 | 0.81 | 0.70 |

*Table 33.* Simple task accuracy on Llama3.1$_{\text{8B}}$.

| Task name | Original | w/ instruction | w/ $\mathcal{M}$ | w/ $\mathcal{M}$ of Llama3.1$_{\text{8B-inst}}$ |
|---|---|---|---|---|
| lowercase_first_letter | 0.00 | 1.00 | 1.00 | 0.15 |
| park-country | 0.06 | 0.78 | 0.75 | 0.12 |
| synonym | 0.04 | 0.42 | 0.49 | 0.10 |
| ag_news | 0.00 | 0.49 | 0.82 | 0.00 |
| word_length | 0.00 | 0.45 | 0.32 | 0.15 |
| present-past | 0.01 | 0.96 | 0.97 | 0.07 |
| capitalize | 0.00 | 1.00 | 0.99 | 0.23 |
| landmark-country | 0.04 | 0.82 | 0.87 | 0.24 |
| english-german | 0.00 | 0.48 | 0.50 | 0.00 |
| sentiment | 0.00 | 0.91 | 0.80 | 0.00 |
| country-capital | 0.05 | 0.89 | 0.85 | 0.04 |
| person-occupation | 0.03 | 0.03 | 0.55 | 0.09 |
| country-currency | 0.00 | 0.06 | 0.61 | 0.00 |
| lowercase_last_letter | 0.00 | 0.93 | 0.81 | 0.00 |
| person-sport | 0.00 | 0.13 | 0.89 | 0.00 |
| person-instrument | 0.00 | 0.15 | 0.68 | 0.01 |
| antonym | 0.13 | 0.66 | 0.66 | 0.11 |
| capitalize_last_letter | 0.00 | 0.76 | 0.22 | 0.00 |
| english-french | 0.00 | 0.73 | 0.80 | 0.02 |
| next_item | 0.02 | 0.67 | 0.85 | 0.21 |
| commonsense_qa | 0.00 | 0.07 | 0.51 | 0.21 |
| singular-plural | 0.02 | 0.95 | 0.96 | 0.07 |
| capitalize_second_letter | 0.00 | 0.44 | 0.22 | 0.00 |
| prev_item | 0.01 | 0.62 | 0.79 | 0.15 |
| capitalize_first_letter | 0.00 | 1.00 | 0.99 | 0.06 |
| english-spanish | 0.00 | 0.71 | 0.78 | 0.00 |
| next_capital_letter | 0.00 | 0.86 | 0.47 | 0.01 |
| national_parks | 0.02 | 0.69 | 0.67 | 0.25 |
| product-company | 0.07 | 0.75 | 0.76 | 0.35 |
| conll2003_organization | 0.06 | 0.64 | 0.48 | 0.28 |
| conll2003_person | 0.07 | 0.89 | 0.86 | 0.36 |
| conll2003_location | 0.07 | 0.79 | 0.79 | 0.37 |
| adjective_v_verb_3 | 0.00 | 0.93 | 1.00 | 0.37 |
| object_v_concept_3 | 0.00 | 0.75 | 0.99 | 0.46 |
| verb_v_adjective_3 | 0.00 | 0.82 | 0.95 | 0.49 |
| fruit_v_animal_3 | 0.00 | 0.99 | 0.97 | 0.43 |
| **Average** | 0.02 | 0.69 | 0.75 | 0.15 |

*Table 34.* Simple task accuracy on Qwen2.5**7B-inst**.

| Task name | Original | w/ instruction | w/ $\mathcal{M}$ | w/ $\mathcal{M}$ of Qwen2.5**7B** |
|---|---|---|---|---|
| lowercase_first_letter | 0.00 | 1.00 | 0.99 | 0.97 |
| park-country | 0.00 | 0.57 | 0.61 | 0.66 |
| synonym | 0.02 | 0.53 | 0.35 | 0.36 |
| ag_news | 0.00 | 0.73 | 0.79 | 0.02 |
| word_length | 0.00 | 0.73 | 0.29 | 0.10 |
| present-past | 0.01 | 0.97 | 0.87 | 0.90 |
| capitalize | 0.17 | 0.99 | 0.99 | 0.92 |
| landmark-country | 0.07 | 0.30 | 0.72 | 0.70 |
| english-german | 0.00 | 0.40 | 0.36 | 0.11 |
| sentiment | 0.03 | 0.95 | 0.92 | 0.53 |
| country-capital | 0.03 | 0.89 | 0.80 | 0.79 |
| person-occupation | 0.09 | 0.04 | 0.49 | 0.38 |
| country-currency | 0.00 | 0.06 | 0.11 | 0.24 |
| lowercase_last_letter | 0.00 | 0.93 | 0.37 | 0.15 |
| person-sport | 0.33 | 0.22 | 0.82 | 0.70 |
| person-instrument | 0.03 | 0.14 | 0.40 | 0.46 |
| antonym | 0.09 | 0.70 | 0.60 | 0.63 |
| capitalize_last_letter | 0.02 | 0.84 | 0.29 | 0.24 |
| english-french | 0.00 | 0.66 | 0.51 | 0.31 |
| next_item | 0.00 | 0.86 | 0.77 | 0.71 |
| commonsense_qa | 0.00 | 0.86 | 0.80 | 0.21 |
| singular-plural | 0.04 | 0.98 | 0.93 | 0.80 |
| capitalize_second_letter | 0.02 | 0.76 | 0.37 | 0.27 |
| prev_item | 0.02 | 0.71 | 0.51 | 0.37 |
| capitalize_first_letter | 0.22 | 1.00 | 1.00 | 0.87 |
| english-spanish | 0.00 | 0.74 | 0.57 | 0.25 |
| next_capital_letter | 0.03 | 0.38 | 0.05 | 0.44 |
| national_parks | 0.03 | 0.41 | 0.65 | 0.58 |
| product-company | 0.14 | 0.43 | 0.73 | 0.55 |
| conll2003_organization | 0.01 | 0.69 | 0.35 | 0.36 |
| conll2003_person | 0.04 | 0.91 | 0.90 | 0.59 |
| conll2003_location | 0.01 | 0.84 | 0.81 | 0.65 |
| adjective_v_verb_3 | 0.00 | 0.99 | 0.84 | 0.70 |
| object_v_concept_3 | 0.00 | 0.97 | 0.98 | 0.78 |
| verb_v_adjective_3 | 0.00 | 0.86 | 0.89 | 0.79 |
| fruit_v_animal_3 | 0.00 | 0.94 | 0.86 | 0.84 |
| **Average** | 0.04 | 0.69 | 0.64 | 0.53 |

*Table 35.* Simple task accuracy on Qwen2.5**7B**.

| Task name | Original | w/ instruction | w/ $\mathcal{M}$ | w/ $\mathcal{M}$ of Qwen2.5**7B-inst** |
|---|---|---|---|---|
| lowercase_first_letter | 0.00 | 1.00 | 0.96 | 0.00 |
| park-country | 0.06 | 0.54 | 0.67 | 0.23 |
| synonym | 0.00 | 0.11 | 0.27 | 0.01 |
| ag_news | 0.00 | 0.75 | 0.02 | 0.01 |
| word_length | 0.02 | 0.70 | 0.11 | 0.00 |
| present-past | 0.02 | 0.94 | 0.94 | 0.08 |
| capitalize | 0.03 | 0.99 | 0.96 | 0.26 |
| landmark-country | 0.06 | 0.33 | 0.71 | 0.26 |
| english-german | 0.00 | 0.36 | 0.21 | 0.00 |
| sentiment | 0.00 | 0.97 | 0.72 | 0.10 |
| country-capital | 0.05 | 0.96 | 0.81 | 0.04 |
| person-occupation | 0.02 | 0.19 | 0.39 | 0.00 |
| country-currency | 0.00 | 0.08 | 0.36 | 0.00 |
| lowercase_last_letter | 0.00 | 0.97 | 0.40 | 0.00 |
| person-sport | 0.06 | 0.39 | 0.68 | 0.05 |
| person-instrument | 0.02 | 0.14 | 0.39 | 0.01 |
| antonym | 0.10 | 0.39 | 0.59 | 0.07 |
| capitalize_last_letter | 0.00 | 0.90 | 0.24 | 0.00 |
| english-french | 0.01 | 0.60 | 0.36 | 0.00 |
| next_item | 0.03 | 0.74 | 0.78 | 0.01 |
| commonsense_qa | 0.00 | 0.50 | 0.77 | 0.60 |
| singular-plural | 0.05 | 0.98 | 0.83 | 0.08 |
| capitalize_second_letter | 0.01 | 0.71 | 0.28 | 0.02 |
| prev_item | 0.02 | 0.52 | 0.48 | 0.01 |
| capitalize_first_letter | 0.05 | 1.00 | 0.94 | 0.01 |
| english-spanish | 0.00 | 0.64 | 0.25 | 0.00 |
| next_capital_letter | 0.00 | 0.75 | 0.66 | 0.03 |
| national_parks | 0.02 | 0.46 | 0.56 | 0.25 |
| product-company | 0.08 | 0.48 | 0.55 | 0.06 |
| conll2003_organization | 0.03 | 0.68 | 0.34 | 0.03 |
| conll2003_person | 0.06 | 0.91 | 0.73 | 0.08 |
| conll2003_location | 0.05 | 0.81 | 0.61 | 0.16 |
| adjective_v_verb_3 | 0.00 | 0.92 | 0.83 | 0.21 |
| object_v_concept_3 | 0.00 | 0.91 | 0.88 | 0.05 |
| verb_v_adjective_3 | 0.00 | 0.86 | 0.82 | 0.20 |
| fruit_v_animal_3 | 0.00 | 0.97 | 0.92 | 0.09 |
| **Average** | 0.02 | 0.68 | 0.58 | 0.07 |

*Table 36.* An example of en-zh translation on different sizes of models. Larger models (Qwen2.5**14B-inst** in this example) tend to generate more explanation words. The attention head mask $\mathcal{M}$ can help the model focus on the task and context words.

| | |
|---|---|
| **Chinese input** | 英格兰人要求必须讲多种语言。 |
| **English gold label** | People in England are required to speak more than one language. |
| **Qwen2.5**14B-inst** with instruction** | The English require that multiple languages be spoken.\n\nHowever, it's important to note that the statement "Englanders require that multiple languages be spoken" is not accurate in a general sense. In reality, English is the predominant language spoken in England, |
| **Qwen2.5**7B-inst** with instruction** | The English demand that multiple languages must be spoken. |
| **Qwen2.5**3B-inst** with instruction** | English people demand that multiple languages must be spoken.\n\nHowever, it seems there might be a slight issue with the phrasing. The correct translation would be:\n\n"English people demand that multiple languages should be spoken."\n\nThis makes more sense as it conveys |
| **Qwen2.5**14B-inst** with $\mathcal{M}$** | The English required that multiple languages be spoken. |
| **Qwen2.5**7B-inst** with $\mathcal{M}$** | The English require that multiple languages be spoken. |
| **Qwen2.5**3B-inst** with $\mathcal{M}$** | The English demand to speak in many languages. |

*Table 37.* Attention head mask similarity between Llama3.1$_{\text{8B-inst}}$ ($\mathcal{M}_{\text{inst}}$) and Llama3.1$_{\text{8B}}$ ($\mathcal{M}_{\text{base}}$).

| Task name | $|\mathcal{M}_{\text{inst}}|$ | $|\mathcal{M}_{\text{base}}|$ | Recall | Jaccard similarity | Random similarity |
|---|---|---|---|---|---|
| en-zh | 832 | 819 | 0.874 | 0.766 | 0.675 |
| en-fr | 860 | 809 | 0.895 | 0.766 | 0.687 |
| en-es | 846 | 811 | 0.906 | 0.797 | 0.679 |
| en-de | 862 | 816 | 0.915 | 0.802 | 0.693 |
| en-ru | 828 | 795 | 0.868 | 0.740 | 0.656 |
| en-ar | 837 | 821 | 0.873 | 0.762 | 0.680 |
| zh-en | 937 | 895 | 0.951 | 0.867 | 0.808 |
| zh-fr | 899 | 843 | 0.943 | 0.839 | 0.739 |
| zh-es | 896 | 856 | 0.921 | 0.817 | 0.747 |
| zh-de | 888 | 834 | 0.928 | 0.816 | 0.724 |
| zh-ru | 909 | 846 | 0.939 | 0.826 | 0.748 |
| zh-ar | 889 | 855 | 0.916 | 0.815 | 0.741 |
| fr-en | 924 | 873 | 0.948 | 0.854 | 0.781 |
| fr-zh | 865 | 821 | 0.912 | 0.799 | 0.699 |
| fr-es | 903 | 833 | 0.929 | 0.805 | 0.734 |
| fr-de | 892 | 827 | 0.931 | 0.811 | 0.721 |
| fr-ru | 899 | 849 | 0.927 | 0.819 | 0.743 |
| fr-ar | 882 | 827 | 0.919 | 0.801 | 0.715 |
| es-en | 925 | 879 | 0.947 | 0.856 | 0.786 |
| es-zh | 863 | 823 | 0.911 | 0.801 | 0.699 |
| es-fr | 909 | 821 | 0.937 | 0.800 | 0.728 |
| es-de | 881 | 825 | 0.921 | 0.803 | 0.712 |
| es-ru | 886 | 835 | 0.905 | 0.783 | 0.724 |
| es-ar | 871 | 839 | 0.911 | 0.808 | 0.716 |
| de-en | 923 | 875 | 0.944 | 0.850 | 0.781 |
| de-zh | 868 | 820 | 0.918 | 0.805 | 0.700 |
| de-fr | 911 | 830 | 0.934 | 0.802 | 0.736 |
| de-es | 920 | 845 | 0.933 | 0.807 | 0.755 |
| de-ru | 893 | 852 | 0.918 | 0.812 | 0.742 |
| de-ar | 885 | 835 | 0.919 | 0.805 | 0.723 |
| ru-en | 923 | 883 | 0.940 | 0.850 | 0.788 |
| ru-zh | 870 | 829 | 0.913 | 0.804 | 0.708 |
| ru-fr | 898 | 836 | 0.933 | 0.818 | 0.732 |
| ru-es | 902 | 831 | 0.930 | 0.805 | 0.731 |
| ru-de | 886 | 838 | 0.930 | 0.824 | 0.726 |
| ru-ar | 873 | 854 | 0.906 | 0.812 | 0.729 |
| ar-en | 943 | 900 | 0.951 | 0.867 | 0.817 |
| ar-zh | 885 | 862 | 0.928 | 0.845 | 0.743 |
| ar-fr | 908 | 843 | 0.926 | 0.805 | 0.745 |
| ar-es | 899 | 834 | 0.935 | 0.818 | 0.732 |
| ar-de | 909 | 859 | 0.951 | 0.859 | 0.758 |
| ar-ru | 888 | 857 | 0.926 | 0.835 | 0.742 |
| lowercase_first_letter | 790 | 672 | 0.835 | 0.623 | 0.549 |
| park-country | 826 | 774 | 0.850 | 0.699 | 0.640 |
| synonym | 833 | 689 | 0.875 | 0.656 | 0.583 |
| ag_news | 737 | 691 | 0.777 | 0.603 | 0.534 |
| word_length | 792 | 673 | 0.798 | 0.579 | 0.551 |
| present-past | 845 | 691 | 0.874 | 0.648 | 0.590 |
| capitalize | 895 | 777 | 0.915 | 0.740 | 0.684 |
| landmark-country | 779 | 685 | 0.848 | 0.658 | 0.553 |
| english-german | 837 | 734 | 0.887 | 0.708 | 0.618 |
| sentiment | 745 | 679 | 0.779 | 0.591 | 0.531 |
| country-capital | 859 | 797 | 0.867 | 0.716 | 0.677 |
| person-occupation | 748 | 688 | 0.814 | 0.639 | 0.538 |
| country-currency | 754 | 744 | 0.806 | 0.668 | 0.577 |
| lowercase_last_letter | 746 | 608 | 0.768 | 0.526 | 0.486 |
| person-sport | 758 | 686 | 0.805 | 0.619 | 0.542 |
| person-instrument | 765 | 646 | 0.837 | 0.622 | 0.520 |
| antonym | 834 | 726 | 0.890 | 0.707 | 0.610 |
| capitalize_last_letter | 752 | 651 | 0.774 | 0.561 | 0.517 |
| english-french | 822 | 727 | 0.867 | 0.686 | 0.604 |
| next_item | 871 | 728 | 0.886 | 0.676 | 0.632 |
| singular-plural | 837 | 704 | 0.874 | 0.664 | 0.596 |
| capitalize_second_letter | 723 | 659 | 0.748 | 0.555 | 0.508 |
| prev_item | 830 | 713 | 0.860 | 0.659 | 0.599 |
| capitalize_first_letter | 774 | 678 | 0.813 | 0.612 | 0.545 |
| english-spanish | 808 | 737 | 0.855 | 0.689 | 0.604 |
| next_capital_letter | 748 | 654 | 0.778 | 0.570 | 0.517 |
| national_parks | 827 | 792 | 0.837 | 0.694 | 0.653 |
| product-company | 754 | 674 | 0.822 | 0.634 | 0.533 |
| conll2003_organization | 826 | 723 | 0.864 | 0.676 | 0.604 |
| conll2003_person | 831 | 710 | 0.887 | 0.692 | 0.597 |
| conll2003_location | 787 | 670 | 0.828 | 0.615 | 0.547 |
| adjective_v_verb_3 | 813 | 756 | 0.868 | 0.719 | 0.620 |
| object_v_concept_3 | 826 | 733 | 0.877 | 0.702 | 0.611 |
| verb_v_adjective_3 | 839 | 768 | 0.867 | 0.708 | 0.644 |
| fruit_v_animal_3 | 827 | 786 | 0.850 | 0.707 | 0.649 |

*Table 38.* Attention head mask similarity between Qwen2.5$_{\text{7B-inst}}$ ($\mathcal{M}_{\text{inst}}$) and Qwen2.5$_{\text{7B}}$ ($\mathcal{M}_{\text{base}}$).

| Task name | $|\mathcal{M}_{\text{inst}}|$ | $|\mathcal{M}_{\text{base}}|$ | Recall | Jaccard similarity | Random similarity |
|---|---|---|---|---|---|
| en-zh | 643 | 611 | 0.894 | 0.771 | 0.666 |
| en-fr | 611 | 587 | 0.853 | 0.719 | 0.618 |
| en-es | 609 | 580 | 0.862 | 0.726 | 0.610 |
| en-de | 602 | 589 | 0.862 | 0.744 | 0.612 |
| en-ru | 614 | 590 | 0.859 | 0.727 | 0.623 |
| en-ar | 618 | 597 | 0.859 | 0.731 | 0.632 |
| zh-en | 666 | 615 | 0.912 | 0.779 | 0.689 |
| zh-fr | 630 | 603 | 0.872 | 0.744 | 0.647 |
| zh-es | 630 | 576 | 0.882 | 0.728 | 0.623 |
| zh-de | 629 | 596 | 0.876 | 0.743 | 0.640 |
| zh-ru | 620 | 590 | 0.871 | 0.739 | 0.628 |
| zh-ar | 619 | 612 | 0.848 | 0.729 | 0.646 |
| fr-en | 668 | 606 | 0.914 | 0.769 | 0.681 |
| fr-zh | 650 | 602 | 0.892 | 0.751 | 0.663 |
| fr-es | 617 | 580 | 0.872 | 0.732 | 0.616 |
| fr-de | 635 | 593 | 0.885 | 0.747 | 0.642 |
| fr-ru | 628 | 590 | 0.864 | 0.720 | 0.634 |
| fr-ar | 625 | 603 | 0.859 | 0.730 | 0.643 |
| es-en | 661 | 618 | 0.892 | 0.757 | 0.687 |
| es-zh | 666 | 607 | 0.914 | 0.773 | 0.681 |
| es-fr | 614 | 583 | 0.864 | 0.727 | 0.617 |
| es-de | 634 | 588 | 0.895 | 0.756 | 0.637 |
| es-ru | 629 | 598 | 0.875 | 0.743 | 0.642 |
| es-ar | 620 | 591 | 0.860 | 0.723 | 0.629 |
| de-en | 680 | 632 | 0.911 | 0.783 | 0.718 |
| de-zh | 669 | 611 | 0.907 | 0.763 | 0.687 |
| de-fr | 640 | 590 | 0.880 | 0.730 | 0.644 |
| de-es | 636 | 594 | 0.884 | 0.745 | 0.644 |
| de-ru | 636 | 611 | 0.871 | 0.744 | 0.660 |
| de-ar | 626 | 609 | 0.862 | 0.739 | 0.649 |
| ru-en | 676 | 625 | 0.917 | 0.787 | 0.707 |
| ru-zh | 670 | 608 | 0.918 | 0.775 | 0.685 |
| ru-fr | 629 | 589 | 0.866 | 0.720 | 0.634 |
| ru-es | 626 | 595 | 0.861 | 0.722 | 0.637 |
| ru-de | 626 | 595 | 0.877 | 0.747 | 0.637 |
| ru-ar | 641 | 605 | 0.868 | 0.728 | 0.658 |
| ar-en | 671 | 652 | 0.911 | 0.815 | 0.729 |
| ar-zh | 677 | 649 | 0.917 | 0.814 | 0.732 |
| ar-fr | 638 | 625 | 0.877 | 0.766 | 0.674 |
| ar-es | 641 | 624 | 0.883 | 0.772 | 0.676 |
| ar-de | 637 | 637 | 0.873 | 0.774 | 0.684 |
| ar-ru | 643 | 636 | 0.876 | 0.771 | 0.689 |
| lowercase_first_letter | 520 | 414 | 0.758 | 0.506 | 0.416 |
| park-country | 602 | 517 | 0.845 | 0.641 | 0.550 |
| synonym | 602 | 502 | 0.861 | 0.643 | 0.536 |
| ag_news | 554 | 486 | 0.782 | 0.576 | 0.493 |
| word_length | 568 | 452 | 0.759 | 0.507 | 0.473 |
| present-past | 577 | 494 | 0.816 | 0.603 | 0.514 |
| capitalize | 626 | 503 | 0.871 | 0.634 | 0.552 |
| landmark-country | 592 | 478 | 0.860 | 0.624 | 0.509 |
| english-german | 581 | 473 | 0.831 | 0.595 | 0.498 |
| sentiment | 565 | 481 | 0.807 | 0.590 | 0.496 |
| country-capital | 615 | 515 | 0.835 | 0.614 | 0.556 |
| person-occupation | 593 | 459 | 0.841 | 0.580 | 0.493 |
| country-currency | 590 | 496 | 0.841 | 0.623 | 0.524 |
| lowercase_last_letter | 516 | 407 | 0.764 | 0.508 | 0.409 |
| person-sport | 569 | 464 | 0.813 | 0.575 | 0.484 |
| person-instrument | 551 | 456 | 0.768 | 0.533 | 0.467 |
| antonym | 602 | 509 | 0.868 | 0.661 | 0.543 |
| capitalize_last_letter | 519 | 432 | 0.750 | 0.517 | 0.430 |
| english-french | 586 | 473 | 0.837 | 0.597 | 0.501 |
| next_item | 600 | 491 | 0.837 | 0.604 | 0.525 |
| singular-plural | 575 | 473 | 0.825 | 0.593 | 0.495 |
| capitalize_second_letter | 542 | 431 | 0.782 | 0.530 | 0.441 |
| prev_item | 603 | 477 | 0.834 | 0.584 | 0.514 |
| capitalize_first_letter | 521 | 433 | 0.760 | 0.526 | 0.432 |
| english-spanish | 589 | 475 | 0.857 | 0.619 | 0.505 |
| next_capital_letter | 531 | 427 | 0.745 | 0.497 | 0.432 |
| national_parks | 587 | 510 | 0.825 | 0.623 | 0.534 |
| product-company | 560 | 445 | 0.820 | 0.570 | 0.463 |
| conll2003_organization | 591 | 501 | 0.846 | 0.635 | 0.529 |
| conll2003_person | 593 | 515 | 0.862 | 0.669 | 0.542 |
| conll2003_location | 584 | 492 | 0.852 | 0.638 | 0.517 |
| adjective_v_verb_3 | 621 | 549 | 0.883 | 0.708 | 0.592 |
| object_v_concept_3 | 635 | 525 | 0.872 | 0.652 | 0.579 |
| verb_v_adjective_3 | 613 | 524 | 0.865 | 0.662 | 0.563 |
| fruit_v_animal_3 | 617 | 562 | 0.867 | 0.704 | 0.600 |

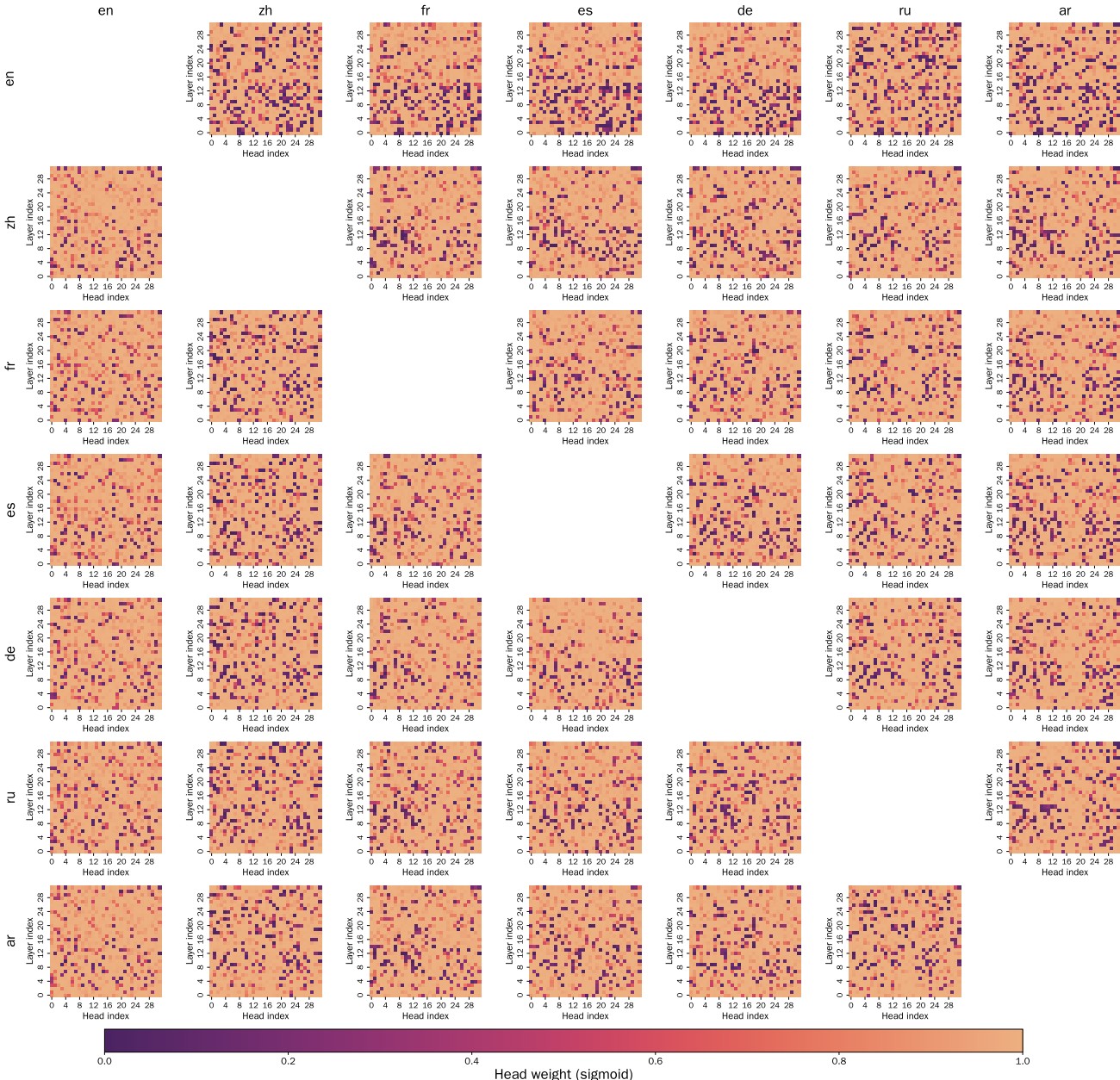

*Figure 7.* Visualization of sigmoid attention head mask weight $\sigma(\mathbf{M})$ for all translation pairs (translate from left labels to top labels) on Llama3.1$_{\text{8B-inst}}$. Heads with a value greater than 0.5 are selected in the final mask $\mathcal{M}$.

## C.3. More Removal Experiments

### C.3.1. ONE-BY-ONE HEAD REMOVAL

In the one-by-one head removal experiment depicted in Figure 2, we unveil the model's translation functionality by removing a certain amount of attention heads. We provide more detailed results to enhance the conclusions that: (1) the model behavior change is not caused by removing certain independent, influential attention heads; (2) the model can maintain its original behavior as long as the functional pathway remains intact, even if the model loses more attention heads.

**Remove by influential order** Figure 2 involves many target token's rank surges when removing certain attention heads, and we claim that removing heads in the order of target token's rank increments caused by each head will not accelerate the switch in model behavior. To verify this point, we select the first 320 heads from the ascending order of the task head weight, and for these 320 heads, we re-order them by the target token's rank increment caused by each head in descending order. Then we remove these heads one by one and compare their token rank changes with the original order. The token rank changes are illustrated in Figure 8. In this influential order, the model could indeed perform higher target token rank in the early stage, but its domination is postponed to around 290 heads. Therefore, removing more influential heads does not necessarily lead to a faster switch in the model behavior.

**Remove on the negative initialized mask** We provide the token ranks and the model outputs of the negative initialized mask in Figure 9. The behavior switch of the model happens at removing the 355th head, which is nearly 200 heads later than the results in Figure 2. Although it loses more attention heads, the model can still maintain certain instruction following behavior before the switch.

### C.3.2. QUERY-HEAD REMOVAL

Equation (2) actually applies $\mathcal{M}$ on the value heads of the MHA layer, and it ignores the attention scores calculated by query and key heads. To explore whether such masks can be applied to the attention scores, we move the mask value inside the softmax function in Equation (1) to make it work on the query (also key) head outputs. In this configuration, masking out a head means that the head pays uniformly equal attention to all the input tokens. Nevertheless, the training of this query-head mask fails to steadily trigger the model functionality on most of the tasks. As described in Section 4, the attention head mask is not only a simple selection of higher or lower attention scores, and even unused heads are still working to focus on important tokens. This can explain why the query-head mask fails to work. Table 39 shows a case study of failed translations.

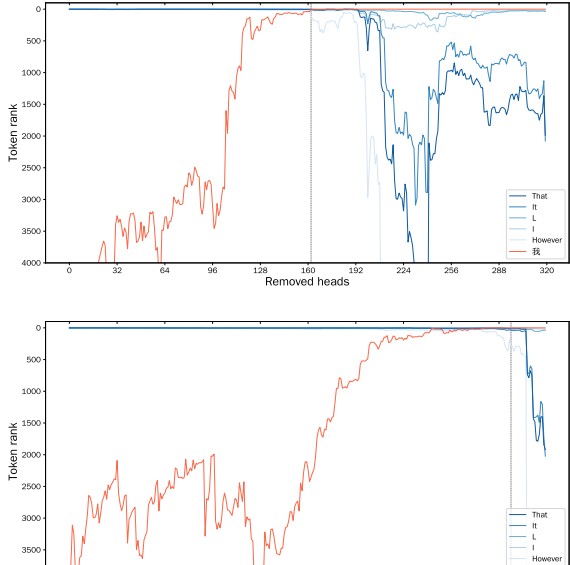

*Figure 8.* Prompt Llama3.1$_{8B\text{-}inst}$ with *"I see a llama sleeping in my backyard."*, and remove the first 320 attention heads from the full model by head weight of en-zh translation task in different orders. We use a gray vertical line to indicate where the target token becomes the top-1 token. **Top:** In original ascending order of head weights. **Bottom:** In descending order of target token's rank increments.

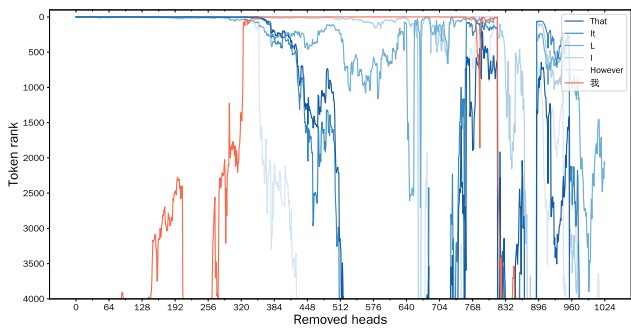

| # of removed | Generated outputs |
|---|---|
| 0 | That's quite a surprise! Llamas are not typically found in backyards, so it's |
| 1 | That's quite a surprise! Llamas are not typically found in backyards, so it's |
| ... | ... |
| 235 | That's a pretty interesting sight! I've seen a llama sleeping in my backyard before, but I |
| 236 | I see a llama sleeping in my backyard. |
| ... | ... |
| 297 | I see a llama sleeping in my backyard. |
| 298 | You're not seeing a llama sleeping in your backyard, you're seeing a llama sleeping in your backyard |
| 312 | That's a pretty surreal scene! A llama sleeping in your backyard! Maybe it's just a visitor |
| 341 | You're looking at a llama that's sleeping in your backyard. |
| 342 | I see a llama sleeping in my backyard. |
| ... | ... |
| 354 | I see a llama sleeping in my backyard. |
| 355 | 我看到一只羊在我的后院睡觉。*(I see a sheep sleeping in my backyard.)* |
| ... | ... |
| 693 | 我看到一只羊在我的后院睡觉。*(I see a sheep sleeping in my backyard.)* |
| 694 | 我看到一只在我家中睡眠。*(I see a sleeping in my house.)* |
| 695 | 我在我的后院看到睡牛。*(I see sleeping ox in my backyard.)* |
| 742 | 我看到一只睡在我的后院的后院。见到一只睡在我的 *(I see a backyard sleeping in my backyard. See sleeping in my)* |
| 743 | I see a llama sleeping in my backyard. |
| 786 | I see a llama sleeping in my backyard.user.sideassistant.help |
| 798 | user's helpmate's helpmate's helper's helper's helper's helper's helper's helper's |
| ... | ... |

*Figure 9.* In the negative initialization scenario, we prompt Llama3.1$_{8B\text{-}inst}$ with *"I see a llama sleeping in my backyard."*, and remove attention heads from the full model by head weight of en-zh translation task in ascending order. Use the same notations as in Figure 2.

*Table 39.* Case study of failed translations in the query-head removal scenario.

| **Case 1:** Fail to translate | |
| --- | --- |
| **Input** | Lots of people from England speak more than one language. |
| **Output** | Many people from England speak more than one language. |

| **Case 2:** Low quality translation | |
| --- | --- |
| **Input** | Niceness is not solely owned by Western tradition. |
| **Output** | 西方传统并不是唯一的好客。 *(Western tradition is not solely hospitable.)* |

| **Case 3:** Mix with other languages | |
| --- | --- |
| **Input** | Due to our population, there are few virgin forests still left in the United States. |
| **Output** | En realidad, debido a nuestra población, todavía quedan pocos bosques virgenes en los Estados Unidos. |
| **Input** | Japan and Sweden are members of the Auld Alliance. |
| **Output** | 日本とスウェデンはオルアライアンスのメンバです。 |

From these cases, we discover an interesting phenomenon that the model may occasionally output translations in other languages (Spanish and Japanese in Case 3), even though it has never encountered these languages during the mask training. This suggests that merely unifying the distribution of attention scores is insufficient for the model to accurately elicit the corresponding functionality. In addition, Section C.2 mentions that the functionality of attention heads is not likely to be concentrated in the KV heads. Combined with the results of query-head removal, we infer that the functionality of attention heads is more likely to be distributed in the output projection matrix $\mathbf{W}_O$, which may serve as a role similar to the FFN layer in the model.

### C.3.3. FFN LAYER REMOVAL

This section mainly explains that removing FFN layers is not necessary for triggering functionalities. In our experiments, we only apply the mask $\mathcal{M}$ on the attention heads, but the mask can also be applied to FFN layers by removing their contributions to the residual stream. Similar to Equation (2), for the input $\mathbf{X}$ at the decoder layer $l$, if we use a binary mask $m_l \in \{0, 1\}$ to indicate whether the FFN layer is selected in the model, then the decoder layer output can be written as:

$$\text{Decoder}_l(\mathbf{X}) = \mathbf{X} + \text{MHA}(\mathbf{X}) + m_l \text{FFN}\left(\mathbf{X} + \text{MHA}(\mathbf{X})\right). \tag{5}$$

*Table 40.* XNLI translation perplexity (PPL) on Llama3.1$_\text{8B-inst}$ when taking FFN layers (32 in total) in the mask. Translation pairs with yellow background use 31 FFN layers in $\mathcal{M}$, others use all 32 FFN layers. Attention heads are also selected in $\mathcal{M}$.

| To
From | **en** | **zh** | **fr** | **es** | **de** | **ru** | **ar** |
| --- | --- | --- | --- | --- | --- | --- | --- |
| **en** | – | 1.700 | 1.130 | 1.221 | 1.330 | 1.915 | 1.869 |
| **zh** | 1.990 | – | 1.537 | 1.701 | 4.143 | 1.828 | 2.523 |
| **fr** | 1.698 | 1.934 | – | 1.437 | 1.543 | 1.936 | 2.042 |
| **es** | 1.509 | 1.970 | 1.283 | – | 1.479 | 1.868 | 2.315 |
| **de** | 3.650 | 4.709 | 1.359 | 1.531 | – | 2.026 | 5.121 |
| **ru** | 2.047 | 2.077 | 1.469 | 3.915 | 4.315 | – | 2.494 |
| **ar** | 2.201 | 2.211 | 1.643 | 4.488 | 5.089 | 1.949 | – |

We take $m_l$ in each layer into the whole mask $\mathcal{M}$, train it together with attention heads on Llama3.1$_\text{8B-inst}$ for XNLI translation tasks, and results are shown in Table 40. After training, most translation tasks still use all the FFN layers. Only 8 tasks remove only one FFN layer in $\mathcal{M}$, and their performance severely degrade compared to the head-only mask in Table 12. Since taking FFN into the mask does not contribute to the study of model functionality, and the model itself tends to utilize all FFN layers, we do not remove the FFN layers in the main experiments.

### C.4. A Probe of Layer Outputs

For Figure 4, we provide a probing example of layer output logits in the vocabulary space in Table 41. We prompt Llama3.1$_\text{8B-inst}$ with *"I see a llama sleeping in my backyard."*, and check the layer outputs in different settings. From the table, we can observe that model outputs w/ instruction and w/ $\mathcal{M}$ exhibit a high degree of overlap in the top-5 tokens in the latter layers of the model (Layer 22–30), reflecting the similarity in information flow when executing the functionality. At the output layer (Layer 31), the masked model produces translations in the target language, as well as semantically aligns other top tokens with the input text. This also demonstrates the semantic preservation characteristic of the translation task.

### C.5. IWSLT2017 Evaluation Examples

In Section 5, we evaluate the BLEU scores of the model with $\alpha$-scaled $\mathcal{M}$ on the IWSLT2017 translation dataset. Considering the style difference between the models shown in Table 36, in the evaluation, we clean up the generated outputs so that they contain no explanation words or punctuation marks. Therefore, the BLEU scores reported in Table 8 are more reliable in reflecting the translation quality of the model. We provide some examples of the model outputs in Table 42. Overall, whether the instructed model includes scaled $\mathcal{M}$ does not make too much difference in the generated sentences. However, the model w/ scaled $\mathcal{M}$ generally provides responses that better align with the labels.

*Table 41.* A probing of the logit outputs in the vocabulary space in Figure 4. White background indicates the original model without instructions, blue background indicates the model w/ instruction, and green background indicates the model w/ $\mathcal{M}$.

| Layer | Top-k tokens | | | | |
|---|---|---|---|---|---|
| | Top1 | Top2 | Top3 | Top4 | Top5 |
| Embedding | illo | abil | incer | otron | câ |
| 0 | | Alam | cheng | Voy | reff |
| | cheng | the | čin | utas | Alam |
| | spun | ouch | ضي | Alam | " |
| 1 | 'gc | | forget | .netbeans | alink |
| | 'gc | alink | extensions | utas | .netbeans |
| | 'gc | extensions | alink | Combination | pok |
| 2 | 'gc | | purch | the | .netbeans |
| | 'gc | .netbeans | -toggler | cheng | |
| | 'gc | edir | utas | : | utar |
| 3 | 'gc | | .CR | edl | .netbeans |
| | 'gc | .netbeans | reff | edn | edl |
| | 'gc | utar | edir | .CR | oğ |
| 4 | 'gc | #ab | #aa | __$ | -toggler |
| | 'gc | -toggler | шиб | ATAB | \Dependency |
| | 'gc | #ab | ên | #aa | edir |
| 5 | #ab | 'gc | #aa | PFN | #ac |
| | #ab | 'gc | #ac | kke | —二 |
| | kke | 'gc | #ab | edir | edl |
| 6 | 'gc | #ad | -*-\r\n | #aa | #af |
| | -*-\r\n | #ad | #ab | .netbeans | 'gc |
| | 'gc | öğ | .netbeans | шиб | kke |
| 7 | LLU | -*-\r\n | chalk | kke | buurt |
| | LLU | -LAST | -*-\r\n | 'gc | #ab |
| | kke | шиб | -*-\r\n | Vectors | .ArgumentParser |
| 8 | #ad | #ab | меть | ibase | morgan |
| | emain | 'gc | #af | chalk | ektor |
| | kke | morgan | Sharper | 'gc | řes |
| 9 | ibase | morgan | #aa | amet | /***/ |
| | poil | .SIG | GetInt | emain | >tag |
| | GetInt | кту | reopen | hub | @js |
| 10 | -*-\r\n | #af | #aa | #ab | |
| | 'gc | >tag | .SIG | цес | poil |
| | 'gc | vertisement | eyim | Sharper | hamster |
| 11 | #aa | #ae | #af | _TAC | )frame |
| | ruz | корист | 감 | >tag | ncy |
| | koli | olik | itom | GetInt | aliz |
| 12 | # | -*-\r\n | _TAC | Sharper | |
| | 'gc | корист | pNet | dime | Sharper |
| | TAC | ouser | lant | gov | ../../../.. |
| 13 | _TAC | # | HITE | #aa | |
| | 'gc | pNet | .Reporting | Sharper | dissolve |
| | cocci | ../../../../ | hale | unset | gov |
| 14 | _TAC | # | -lfs | -wsj | (DBG |
| | -wsj | ContentLoaded | � | меть | pNet |
| | rych | � | bette | _TAC | esan |
| 15 | -wsj | TAC | .DO | aken | atre |
| | -wsj | >tag | :uint | lap | hon |
| | ContentSize | grese | ../../../.. | rych | acz |
| 16 | styleType | =ア | -wsj | zsche | #af |
| | RetVal | hon | )frame | -wsj | Macy |
| | zsche | itsu | ContentSize | hend | cocci |
| 17 | unfortunately | trom | itious | clid | ORY |
| | hon | artz | increasingly | confidently | couch |
| | hend | cocci | esan | owy | |
| 18 | unfortunately | WEBPACK | Levy | ORY | HS |
| | confidently | bes | WSTR | increasingly | hon |
| | owy | enth | hek | :UIAlert | bes |
| 19 | sounds | unfortunately | Highland | probably | unlikely |
| | hon | bes | increasingly | | ( |
| | wonderfully | bes | | �单 | otive |
| 20 | That | sounds | likelihood | | unlikely |
| | hon | bes | in | increasingly | MainAxisAlignment |
| | a | | beg | I | bes |
| 21 | That | That | chances | likelihood | zcze |
| | I | my | in | c | p |
| | I | ..\n | a | it | |
| 22 | That | That | chances | -lnd | likelihood |
| | I | my | you | me | c |
| | I | ..\n | a | my | |
| 23 | it | That | It | unless | |
| | I | you | my | in | ..\n |
| | ..\n | I | a | in | |
| 24 | it | while | It | While | That |
| | my | I | ..\n | in | you |
| | ..\n | I | …\n | a | my |
| 25 | it | That | that | while | That |
| | my | ..\n | I | in | you |
| | ..\n | I | …\n | in | in |
| 26 | That | that | That | THAT | it |
| | my | I | in | ..\n | you |
| | ..\n | I | in | in | …\n |
| 27 | that | That | That | THAT | that |
| | in | you | my | ..\n | …\n |
| | ..\n | I | in | in | in |
| 28 | that | That | That | ..\n | THAT |
| | …\n | ..\n | in | /stdc | backyard |
| | …\n | ..\n | in | in | a |
| 29 | That | that | That | ..\n | exempt |
| | …\n | /stdc | in | ..\n | _exempt |
| | …\n | ..\n | in | /stdc | واء |
| 30 | That | It | I | L | That |
| | backyard | _exempt | 我 | in | ..\n |
| | 我 | Un | I | I | A |
| 31 | That | It | L | I | However |
| | 我 | 你 | 您 | 在 | 有 |
| | 我 | I | A | 一个 | It |

*Table 42.* Examples of IWSLT2017 en-zh translation on instructed Llama3.1$_{\text{8B-inst}}$ and the instructed model with $\alpha$-scaled $\mathcal{M}$. We provide their BLEU scores in the bracket, and highlight the alignments and differences among the translations.

| | |
|---|---|
| **Chinese input** | 而有的时候，一个小小的样板模型的经验会帮助我们将"糟了"的那一刻变成了""哈哈"的一刻 |
| **English gold label** | And sometimes a little prototype of this experience is all that it takes to turn us from an uhoh moment to a tada moment |
| **Instructed** *(0.556)* | And sometimes a small models experience will help us turn a oh no moment into a ha ha moment |
| **w/ scaled $\mathcal{M}$** *(0.640)* | And sometimes a small prototype models experience will help us turn a oh no moment into a ha ha moment |
| **Chinese input** | 我在想你们会怎么选，近来我问了我的朋友很多次这个问题，他们的回答都是"回到过去"。 |
| **English gold label** | And I wonder what youd choose because Ive been asking my friends this question a lot lately and they all want to go back |
| **Instructed** *(0.616)* | Ive been asking my friends this question a lot lately and they all say go back in time |
| **w/ scaled $\mathcal{M}$** *(0.813)* | Ive been wondering how you would choose and lately Ive asked my friends this question a lot and they all say go back in time |
| **Chinese input** | 大多数人建的塔的平均高度是20英尺 商学院学生，大概是一半 律师好一些，但也好不到哪里去 幼儿园的孩子，比大多数的成年人要好 |
| **English gold label** | So the average for most people is around 20 inches business schools students about half of that lawyers a little better but not much better than that kindergarteners better than most adults |
| **Instructed** *(0.749)* | Most people build towers to an average height of 20 feet Business school students are probably about half as good as lawyers but not much better Preschool children are better than most adults |
| **w/ scaled $\mathcal{M}$** *(0.720)* | Most people build towers with an average height of 20 feet The law students were about half as good but not much better The kindergarten children were better than most adults |

## C.6. Experiments on MoE Model

MoE (Mixture-of-Experts) language models divide the model's FFN (Feed-Forward Network) layer into multiple experts, and during inference, only a subset of these experts is selected by routers to contribute to the computation. MoE models often achieves better performance than dense models with a similar number of activated parameters and is gradually becoming a popular research topic. In this section, we conduct experiments to verify whether the attention head mask can still switch the model's behavior in MoE models.

We select Phi-3.5-MoE-instruct (AdinaTru, 2024) (16×3.8B parameters, with 6.6B active parameters when using 2 experts) for head mask training on the English and Chinese XNLI translation task. As shown in Table 43-44, we observe that while head masking in the MoE model could trigger behavior switch, it does not lead to performance improvements on the generative metric. This might be because the majority of MoE model parameters are concentrated in the FFN layers, while the attention layer accounts for only a small portion of parameters. We propose another two possible explanations: (1) The primary functionality of MoE might reside in the FFN, with the router playing a role similar to attention head masking for function selection. (2) The task-level mask and the token-level router may not work well together. Improving the effectiveness of head masks in MoE models is an important direction for our future research.

*Table 43.* XNLI translation perplexity (PPL) on Phi-3.5-MoE$_{inst}$.

| w/ instruction | | | | | | |
|---|---|---|---|---|---|---|
| To
From | **en** | **zh** | **fr** | **es** | **de** | **ru** | **ar** |
| **en** | – | 1.088 | 1.046 | 0.857 | 1.008 | 1.286 | 0.851 |
| **zh** | 2.747 | – | 2.160 | 2.371 | 2.317 | 1.825 | 1.237 |
| w/ $\mathcal{M}$ | | | | | | |
| To
From | **en** | **zh** | **fr** | **es** | **de** | **ru** | **ar** |
| **en** | – | 0.786 | 0.684 | 0.558 | 0.680 | 0.877 | 0.558 |
| **zh** | 1.098 | – | 1.048 | 1.141 | 1.202 | 1.148 | 0.746 |

*Table 44.* XNLI translation ROUGE-L on Phi-3.5-MoE$_{inst}$.

| w/ instruction | | | | | | |
|---|---|---|---|---|---|---|
| To
From | **en** | **zh** | **fr** | **es** | **de** | **ru** | **ar** |
| **en** | – | 0.448 | 0.650 | 0.674 | 0.628 | 0.429 | 0.425 |
| **zh** | 0.519 | – | 0.465 | 0.484 | 0.449 | 0.378 | 0.344 |
| w/ $\mathcal{M}$ | | | | | | |
| To
From | **en** | **zh** | **fr** | **es** | **de** | **ru** | **ar** |
| **en** | – | 0.400 | 0.361 | 0.568 | 0.481 | 0.282 | 0.278 |
| **zh** | 0.558 | – | 0.314 | 0.327 | 0.350 | 0.231 | 0.237 |

