# OpenReview forum: "Heads up! Large Language Models Can Perform Tasks Without Your Instruction via Selective Attention Head Masking"
_ICML.cc/2025/Conference — ICML 2025 poster_

### Official Review · Reviewer_7BU1 · 2025-03-04

**Overall Recommendation:** 4

**Summary:**

This article explores the ability of large language models (LLMs) to perform tasks without relying on explicit instructions through selective attention head masks. It is found that there exists a "functional path" composed of attention head combinations within the model, which is crucial for task execution. Experiments have shown that masking strategies can effectively improve task performance, and pre trained models already have task related knowledge that can be triggered by masking. The study proposes a method for detecting functional pathways, revealing the key role of attention head combinations and providing a new perspective for understanding the internal mechanisms of LLMs.

**Claims And Evidence:**

Yes.

**Essential References Not Discussed:**

No.

**Experimental Designs Or Analyses:**

Yes.

**Methods And Evaluation Criteria:**

Yes.

**Other Comments Or Suggestions:**

No.

**Other Strengths And Weaknesses:**

**Strengths**

1. Novel Insight into Model Functionality. The paper  revealing how specific combinations of attention heads enable task execution without explicit instructions.

2. Practical Methodology. The proposed attention head masking technique is simple yet effective, requiring no modification to model parameters. This approach can be easily applied across various models and tasks

**Weaknesses**

1. Lack of rigorous theoretical analysis.

2. There are no larger scale experiments. This article states that "the Large model shows better head functionality", but the maximum is only the 14B.

**Questions For Authors:**

1. Can you conduct corresponding experiments on a larger scale, such as on 72B. And conduct experiments on the MOE model. I am not sure if the MOE model has similar improvements as the dense model. Because generally speaking, the MOE model may be more inclined towards memory, and the proportion of attention in the MOE model is relatively small.

2. Another curious question for me is whether this method can directly stimulate the reasoning ability of the base model. For example, for deepseek v3, masking some of  heads can significantly indicate its performance on benchmarks like AIM. If masking attention can stimulate better reasoning ability, I believe this will be a significant contribution.

**Relation To Broader Scientific Literature:**

This article establishes a connection between instruction tuning and pruning of attention heads, which is a very interesting work. This provides a new perspective for us to better understand the essence of instruction tuning and the internal mechanisms of LLM.

**Theoretical Claims:**

No. There is no  theoretical claim.

---

> ### Author Rebuttal · Authors · 2025-04-01
>
> Many thanks for your appreciation and valuable remarks on our work. We would like to answer and clarify your concerns as below.
>
> > W1. Theoretical analysis
>
> Since our conclusions are primarily derived from experiments and observations rather than strict theoretical analysis, we sincerely apologize for the shortcomings in this regard. Nevertheless, some studies [1][2] have provided insights that support the validity of our approach, in which we regard attention heads as computational nodes in the information flow and analyze their functionality.
>
> [1] Yao, Y., Zhang, N., Xi, Z., Wang, M., Xu, Z., Deng, S., & Chen, H. (2024). Knowledge circuits in pretrained transformers. arXiv preprint arXiv:2405.17969.
> [2] Hao, Y., Dong, L., Wei, F., & Xu, K. (2021, May). Self-attention attribution: Interpreting information interactions inside transformer. In Proceedings of the AAAI Conference on Artificial Intelligence (Vol. 35, No. 14, pp. 12963-12971).
>
> > W2. Larger scale experiments
>
> In Sec. 3.4, we stated that "Larger model shows better head functionality," primarily based on the criterion of whether *behavior switching* can be achieved. This is because, in models smaller than 3B, masking out certain attention heads does not reliably trigger behavior switching. To further support this claim, we conducted additional experiments on Qwen2.5-32B-Instruct and found that it also exhibits behavior switching in translation tasks (see Q1).
>
> > Q1. Large model & MoE model
>
> We are very sorry, but we currently do not have the resources to conduct experiments on a model as large as 72B. However, we have performed training on a 32B model (Qwen2.5-32B-Instruct). Due to time constraints, we only trained head masks for en to other six languages translation task. Generally, we found that applying masks can still improve model performance on the 32B model.
>
> - PPL
>
> ||en-zh|en-fr|en-es|en-de|en-ru|en-ar|
> |-|-|-|-|-|-|-|
> |with instruction|2.739|1.325|1.385|1.448|1.587|1.960|
> |with mask|**0.943**|**0.553**|**0.448**|**0.554**|**0.832**|**1.326**|
>
> - ROUGE-L
>
> ||en-zh|en-fr|en-es|en-de|en-ru|en-ar|
> |-|-|-|-|-|-|-|
> |with instruction|0.470|0.428|0.415|0.445|0.388|**0.293**|
> |with mask|**0.684**|**0.703**|**0.752**|**0.708**|**0.449**|0.290|
>
> For MoE models, we selected Phi-3.5-MoE-instruct (16×3.8B, activating 6.6B) for head mask training on the en-zh translation task. We observed that while head masking in the MoE model could trigger behavior switch, it did not lead to performance improvements. This might be because MoE model parameters are more inclined toward memory. We propose another two possible explanations: **(1)** The primary functionality of MoE might reside in the FFN, with the router playing a role similar to attention head masking for function selection. **(2)** The task-level mask and the token-level router may not work well together. Improving the effectiveness of head masks in MoE models is an important direction for our future research.
>
> - PPL
>
> ||en-zh|en-fr|en-es|en-de|en-ru|en-ar|zh-en|zh-fr|zh-es|zh-de|zh-ru|zh-ar|
> |-|-|-|-|-|-|-|-|-|-|-|-|-|
> |with instruction|1.088|1.046|0.857|1.008|1.286|0.851|2.747|2.160|2.371|2.317|1.825|1.237
> |with mask|**0.786**|**0.684**|**0.558**|**0.680**|**0.877**|**0.558**|**1.098**|**1.048**|**1.141**|**1.202**|**1.148**|**0.746**|
>
> - ROUGE-L
>
> ||en-zh|en-fr|en-es|en-de|en-ru|en-ar|zh-en|zh-fr|zh-es|zh-de|zh-ru|zh-ar|
> |-|-|-|-|-|-|-|-|-|-|-|-|-|
> |with instruction|**0.448**|**0.650**|**0.674**|**0.628**|**0.429**|**0.425**|0.519|**0.465**|**0.484**|**0.449**|**0.378**|**0.344**|
> |with mask|0.400|0.361|0.568|0.481|0.282|0.278|**0.558**|0.314|0.327|0.350|0.231|0.237
>
> > Q2. Stimulate reasoning ability
>
> After the release of DeepSeek-R1, we attempted to use R1 to generate high-quality reasoning chains for simple tasks and then trained head masks on these chains in an effort to activate reasoning capabilities in other instruct models. Unfortunately, our results were not as promising as we had hoped. Whether we applied the mask directly or after explicitly prompting the model to "think step by step," the model was still unable to achieve high-quality reasoning solely by selecting specific attention heads. (It may still require parameter modification to stimulate its potential reasoning capabilities, such as LIMO[3])
>
> For the masks we obtained, they tended to discard only a small number of attention heads (less than 10% in a 7B model), and for complex tasks like mathematics, almost all attention heads were retained. This leads us to believe that reasoning is a complex process involving the coordination of multiple functions at different stages of generation. As a result, it may not be feasible to activate reasoning abilities solely through head masking at this level of "task granularity".
>
> [3] Ye, Y., Huang, Z., Xiao, Y., Chern, E., Xia, S., & Liu, P. (2025). LIMO: Less is More for Reasoning. arXiv preprint arXiv:2502.03387.

---

> > ### Comment · Reviewer_7BU1 · 2025-04-04
> >
> > Thanks for the author's response. I understand that some experiments may  be difficult to implement due to lack of resources. I maintain my positive score, and recommend the author to include the results of MOE model in the manuscript.

---

> > > ### Author Response · Authors · 2025-04-07
> > >
> > > Dear Reviewer 7BU1,
> > >
> > > Thank you for your understanding regarding our limited resources. And we will include a discussion of the MoE model results in the next version.
> > >
> > > In addition, although masking attention heads does not seem to stimulate the model’s reasoning abilities, this phenomenon has still shown effectiveness in other scenarios, such as in multimodal models (e.g. speech model SALMONN, and corresponding speech-related tasks) [[Results🔗]](https://i.imgur.com/rWAUytC.png) and in few-shot ICL scenarios [[Results🔗]](https://i.imgur.com/gvKuCU4.png). You may refer to these links for more details if you are interested.
> > >
> > > **[Update]** In our recent study, we found that applying a similar 0-1 mask to the experts in the MoE model (Deepseek-MoE-16B) can also enable the model to directly perform tasks without explicit instructions. As our approach involves modifying the expert router, this can be viewed as a form of selective masking at the FFN level and at the token granularity. While further investigation is left for future work, this suggests that the masking idea proposed in our paper can also be extendable to expert FFNs to probe their functionalities.

---

### Official Review · Reviewer_uwMV · 2025-03-13

**Overall Recommendation:** 3

**Summary:**

This paper proposes a simple yet effective attention head masking method for large language models. Specifically, it trains a head weight which can indicate the importance of heads to the task. After training, we can map the trained head weights to head mask and use it as the final mask for inference. Moreover, this paper finds that there are structured groups of interdependent attention heads for different tasks in LLMs.

**Claims And Evidence:**

Yes, the claims made in the submission are supported by clear evidence.

**Essential References Not Discussed:**

The related works mentioned in this paper are good and enough to understanding the key contributions of the paper.

**Experimental Designs Or Analyses:**

Yes, the main experimental and ablation study designs and analysis are valid.

**Methods And Evaluation Criteria:**

Yes, the proposed methods and evaluation criteria make sense.

**Other Comments Or Suggestions:**

Please see the above sections.

**Other Strengths And Weaknesses:**

**Strengths**:
1. The paper is easy to follow and understand, and the structure is good.
2. The attention head masking strategy proposed in this paper performs well one various tasks.
3. This work proves that LLMs inherently encode structured functional pathways, where specific attention heads are crucial for executing distinct tasks.

**Weaknesses**:
1. **Training Overhead**: For different model series and different tasks, we need to train different attention head masks, which introduces non-negligible computational overhead. Moreover, the paper is supposed to clarify the training overhead for head weights.
2. **Lack of Experiments**: For the translation tasks, the 100 samples are selected as evaluation set without any motivation and explanations. The paper is supposed to clarify the reason why they choose 100 samples. Moreover, More evaluations on common down-stream tasks like Hellaswag or ARC-C are suggested to include.
3. **Efficiency Analysis**: The efficiency analysis for inference stage is recommended to include in the paper, such as end-to-end latency, or GPU memory comparison; otherwise, there remain concerns about its actual efficiency in real world usage.

**Questions For Authors:**

1. Have you checked the patterns of attention heads selected for different tasks? Can we find some common patterns from those heads?
2. What will the performance of the proposed attention head masking strategy be on few-shot learning?
3. Could you please discuss and compare the findings in your work with the findings in DuoAttention [1]?
4. Following Q3, will your strategy still be effective for long-context scenarios?
5. How to control the masking ratio among all attention heads? Could you please provide some insights or solutions?


[1] Xiao, G., Tang, J., Zuo, J., Guo, J., Yang, S., Tang, H., Fu, Y., & Han, S. (2024). DuoAttention: Efficient long-context LLM inference with retrieval and streaming heads. arXiv. https://arxiv.org/abs/2410.10819

**Relation To Broader Scientific Literature:**

This work provides a deep and comprehensive analysis for uncovering the functional behaviors of attention heads in LLMs. Unlike prior efforts that relied on pruning or external prompts, this study shows that selectively activating relevant heads can improve performance, offering insights into efficiency and controllability.

**Theoretical Claims:**

There is no theoretical claims.

---

> ### Author Rebuttal · Authors · 2025-04-01
>
> Thank you for your thoughtful comments and suggestions on our work. We would like to answer and clarify your concerns as below. As the text length is constricted, we provide the results in the anonymous github (rebuttal.pdf).
>
> First, we would like to clarify that the primary focus of this paper is on the interpretability and functionality of LLMs - specifically, the phenomenon that "tasks can be directly executed by selecting attention heads". We find this phenomenon very interesting, and most of the paper is devoted to validating its generality and attempting to explain its underlying mechanisms. (We have even observed the same phenomenon even in multimodal speech models! See our response to b411) Although using masks can indeed enhance model performance in some cases, we sincerely hope that you will pay more attention to our contribution regarding the interpretability of model functionality.
>
> > W1. Training Overhead
>
> It is indeed unavoidable to train a mask for each task on every model to identify its functional heads. However, because our training only involves trainable parameters equal to the number of attention heads, the process is both fast and requires little memory (only 44 minutes and 16.5GB of VRAM on llama3.1-8b). We have also compared this approach with full fine-tuning and LoRA fine-tuning in terms of resource consumption. (Note that since fine-tuning modifies the model parameters, these two methods would undoubtedly achieve better performance. Nevertheless, we are not proposing a PEFT method, and we have neither modified nor added to the model’s original parameters.)
>
> > W2. Lack of Experiments
>
> Initially, we chose to test on 100 samples because we wanted to use more training samples to locate the attention heads required for the translation task more accurately, and the evaluation set was just used to verify whether the model can perform translation. But, indeed, it's our bad to use such a small number of samples for the quantitative evaluation. As it is on the XNLI dataset, we nonetheless evaluated on the full IWSLT2017 evaluation set in the paper (see Table 7 and App. A) without incorporating any additional training data.
>
> We would also like to argue that with a smaller train-test ratio, we can still achieve results similar to those reported in the paper. Please refer to the experimental results obtained after re-dividing the dataset with an 8:2 ratio.
>
> For other benchmarks: although most benchmark tasks include explicit "instructions" such as options, we have still managed to conduct evaluations on some datasets. We achieved better performance on winogrande and hellaswag.
>
> > W3. Efficiency Analysis
>
> Taking the llama3.1-8b model and the en-zh translation task as an example, we masked out 192 (18.75%) attention heads, and pruned the corresponding Q-head and $W_O$ matrices. Since the attention heads account for only 16.7% of the model’s total parameters, and the pruning only affects Q and O, we actually only saved 2.5% of the parameters. As some studies focusing on pruning can remove 20-30% of the parameters with only minimal performance loss, we would like to clarify again that our work is not aimed at achieving inference efficiency, but rather at locating and interpreting the model’s modularity.
>
> > Q1. Head Pattern
>
> In App. C.2 and Figure 7 we have some discussions on the head patterns, and you may check it out. We also have some other findings: **1.** Similar tasks tend to share more attention heads， for example, 42 translation tasks share 177 common attention heads, but 35 simple tasks share only 5 heads. **2.** Due to sparsity, as the model size increases (from Qwen2.5-1.5B to 14B), each attention head tends to be responsible for fewer tasks on average.
>
> > Q2. Performance on few-shot learning
>
> We conducted some supplementary experiments on simple tasks; please take a look at the results. Summary:
>
> 1. When using the mask obtained from 0-shot training for 5-shot inference, the performance ranking is: instructed 5-shot > mask 0-shot > mask 5-shot > 5-shot.
> 2. For the tasks in Table 2, using a mask trained with 5-shot and then performing mask 5-shot inference yields better performance than instructed 5-shot.
> 3. We combined multiple simple task datasets into a hybrid dataset, where each sample may come from different tasks but includes 5-shot examples from its corresponding task. After training the mask on this hybrid dataset, the model was able to enhance its 5-shot context learning performance (including tasks that were not seen during training).
>
> For our more response please refer to the anobymous github. Sorry for the inconvience!

---

### Official Review · Reviewer_b411 · 2025-03-18

**Overall Recommendation:** 2

**Summary:**

The authors study the case that there are several attention heads if switched off, the models actively can be tuned to perform a specific task without fine-tuning.

**Claims And Evidence:**

- Switching off attention heads leads to similar results as prompting (experiments for language translation)

- Masks can be learned, similar to fine-tuning

**Essential References Not Discussed:**

Check broader scientific literature

**Experimental Designs Or Analyses:**

- Yes, checked the translation experiments and comparison with random selection

**Methods And Evaluation Criteria:**

- Used language translation datasets

**Other Comments Or Suggestions:**

See strengths and weakness section

**Other Strengths And Weaknesses:**

- The method is quite simple and easy to implement

- It is unclear if there are any practical benefits of this method.

- It is unclear, how general performance of the model degrades.

**Questions For Authors:**

- Can authors compare how original performance degrades when removing attention heads
- Can authors perform a comparison with activations with Prompting experiments

**Relation To Broader Scientific Literature:**

There have been other works like https://arxiv.org/abs/2410.05603, which have studied that models are superpositioning multiple tasks at the same time. Please comment how your findings are different.

**Theoretical Claims:**

No theoratical claims

---

> ### Author Rebuttal · Authors · 2025-04-01
>
> Thank your for your revisions and comments. We would provide our clarifications and responses below.
>
> First and foremost, we would like to clarify certain statements in your Summary, Methods and Experimental Designs sections. Our experiments are not limited to language translation tasks; they also include 35 simple tasks (Lines 166-169, left column; Table 2, 4, 6, 9, etc.), which account for nearly half of all our experiments. However, this aspect was not mentioned in the review. Additionally, as a paper focused on the functionality and interpretability of LLMs, we have dedicated substantial portions of the text to exploring and analyzing the mechanism that how model "switches behavior through selective attention heads". While it is true that using head masking can improve performance in certain scenarios, we would greatly appreciate it if you could focus more on the paper’s contributions to model functionality and interpretability.
>
> If there are any aspects of the paper that remain unclear, we are always here to provide further clarification and assistance.
>
> > W1. Benefits
>
> As stated above, our paper primarily contributes to the interpretability of models. We find the phenomenon of "enabling the model to perform tasks without instructions by selecting attention heads" to be very intriguing. We propose an efficient method, applicable to a wide range of tasks, for identifying which attention heads are crucial for the task. This phenomenon is not only observable across various architectures and sizes of LLMs (Section 3), but also applies to multimodal models, including those for speech (see below). These attention heads form interdependent "functional pathways" responsible for executing corresponding functions, thus helping us better understand how model modularity manifests in attention heads (Section 4). Additionally, by selecting these attention heads, the model often achieves a certain level of performance improvement (Section 5).
>
> - Speech model: test on SALMONN
>
> | Task | no instruction | with instruction | with mask |
> | -- | -- | -- | -- |
> | Gender recognition | 0.00 | 97.71 | **97.75** |
> | Emotion recognition | 0.00 | 69.70 | **72.36** |
> | ASV | 0.00 | **93.61** | 93.23 |
> | Audio captioning | 14.90 | 20.89 | **24.33** |
> | S2TT | 15.14 | **34.48** | 33.82 |
>
> > W2/Q1. Degrades
>
> After using the mask, the model undergoes a behavioral change and can no longer perform the original task as intended. As a result, the general performance will inevitably decrease. However, since the mask does not modify or add/remove any of the model’s parameters, we can simply stop using the mask to restore the model's original instruction capabilities. This process is straightforward and flexible.
>
> We also conducted some mask experiments on general benchmarks. Please see our response to uwMV.
>
> > Q2. activation experiments
>
> In the paper, we conducted an experiment (Line 370, right column) to investigate the relationship between attention head activation (focus on special tokens) and the mask. Our findings suggest that the correlation between them is not strong. We discuss this phenomenon further in Appendix C.3.2, where we suggest that the "attentive function" of the attention heads does not play a major role in their selection.

---

### Decision · Program_Chairs · 2025-05-01

**Decision:**

Accept (poster)

**Comment:**

This paper investigates how LLMs can perform specific tasks without explicit instructions through selective attention head masking. The authors demonstrate that by applying discrete 0-1 masks to attention heads, models can directly exhibit task-specific functionalities without modifying any parameters. This phenomenon was validated across various model architectures, including both language and multimodal models, and across diverse tasks such as classification, extraction, and few-shot learning. The reviews recognized several strengths of this work: its novel approach to uncovering functional behaviors of attention heads in LLMs, clear paper structure with easy-to-follow methodology, and the effective performance improvements achieved through the proposed masking strategy. While there were some initial concerns about practical benefits and impact on general model performance, the authors provided additional experiments addressing these issues, including tests on larger models and MoE architectures. Multiple reviewers provided positive evaluations after reviewing the authors' thorough rebuttal. Given the thorough experimental validation, novel insights into model interpretability, and satisfactory author responses to reviewer concerns, I recommend accepting this paper for publication at ICML 2025.